# Unleashing Perception-Time Scaling to Multimodal Reasoning Models

**Yifan Li**[1,2,*]**, Zhenghao Chen**[3,*]**, Ziheng Wu**[3,*]**, Kun Zhou**[4]**, Ruipu Luo**[3]**, Can Zhang**[3]**,
**Zhentao He**[3]**, Yufei Zhan**[5]**, Wayne Xin Zhao**[1,2,†]**, Minghui Qiu**[3,†]

[1]Gaoling School of Artificial Intelligence, Renmin University of China
[2]Beijing Key Laboratory of Research on Large Models and Intelligent Governance
[3]ByteDance [4]School of Information, Renmin University of China
[5]Institute of Automation, Chinese Academy of Sciences

## Abstract

Recent advances in inference-time scaling, particularly those leveraging reinforcement learning with verifiable rewards, have substantially enhanced the reasoning capabilities of Large Vision-Language Models (LVLMs). Inspired by this success, similar strategies have been applied to multimodal reasoning, yet their impact on visual perception remains unclear. To investigate this gap, we introduce DisTANCE, a perception-centric benchmark for visual estimation tasks. Evaluation results show that LVLMs exhibit limited estimation precision, and inference-time scaling offers only marginal gains. We attribute this to the fast perception paradigm of current LVLMs, where visual understanding is treated as a one-shot output without modeling the underlying perceptual process. To address this, we propose Perception-Time Scaling (PTS), a novel paradigm that encourages token-rich perception and decomposes complex perception problems into intermediate tractable sub-problems, thereby enabling perception to align with and benefit from inference-time scaling. Combined with reinforcement learning techniques, PTS significantly improves perception accuracy, raising high-precision performance on DisTANCE from 8.0% to 64.7%, and generalizes well to out-of-domain tasks. Surprisingly, even though PTS data are purely synthetic, combining them with math reasoning data yields consistent gains in both reasoning and real-world perception benchmarks. Further analysis reveals that PTS introduces more perception-related tokens and increases the model's attention to image tokens. Our code and data are released at https://github.com/RUCAIBox/PTS.

## 1 Introduction

Recently, reinforcement learning with verifiable rewards (RLVR) (Guo et al., 2025) has emerged as a popular paradigm, as it enables models to produce longer chains of thought and thereby improves their reasoning abilities. Such approaches are commonly referred to as inference-time scaling methods (Zhao et al., 2023). Inspired by this success, recent work has extended similar strategies to Large Vision-Language Models (LVLMs), yielding notable gains on challenging multimodal reasoning tasks, such as mathematical and multi-disciplinary domains (Wang et al., 2024; Yue et al., 2024). These approaches encourage models to produce longer and more deliberate reasoning chains. However, their benefits appear largely restricted to the reasoning stage, and it remains unclear whether such gains transfer to perception. Indeed, recent studies suggest that reasoning LVLMs may even be more prone to hallucinations (Liu et al., 2025a; Yao et al., 2025).

To systematically examine whether perception can also benefit from inference-time scaling, we introduce a perception-centric benchmark, **DisTANCE** (Distance reasoning Task with Analytical Numeric and Comparative Estimation). DisTANCE comprises synthetic geometric images (*e.g.,* circles, triangles) paired with questions that require visual estimation (*e.g.,* length, perimeter, area).

---

[*] Equal contribution.
[†] Corresponding author.

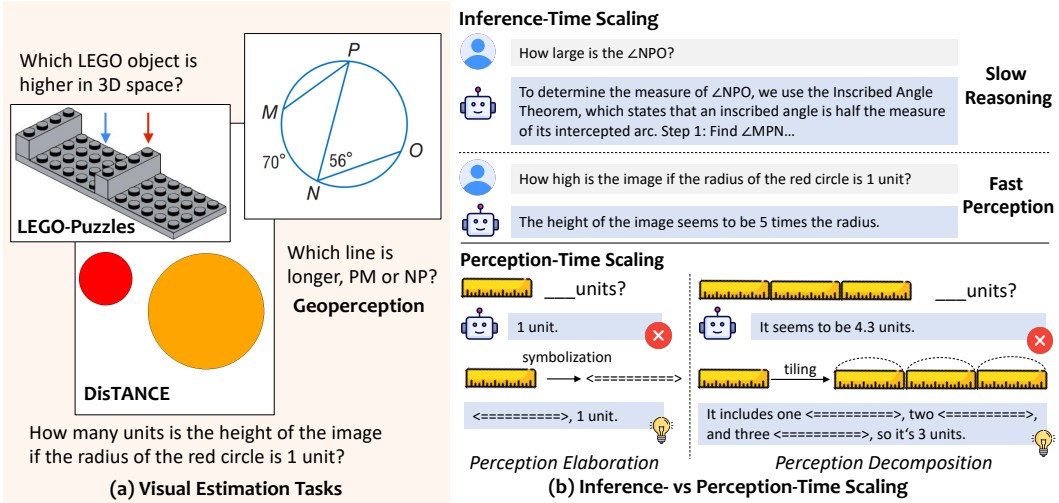

Figure 1: Overview of the visual estimation tasks and comparison between inference-and the proposed perception-time scaling paradigms.

Evaluation results demonstrate that DisTANCE poses substantial challenges for existing LVLMs, while inference-time scaling methods yield only marginal gains over base models. We attribute this limitation to the prevailing *Fast Perception* paradigm of LVLMs (Figure 1), in which perception is treated as a one-shot judgment rather than a process involving intermediate steps. Further analysis reveals that perception-related tokens constitute only a small fraction of the reasoning chain, and the perception accuracy drops noticeably as task complexity grows.

To tackle this limitation, we propose a new paradigm, **Perception-Time Scaling** (PTS), which reformulates perception as a structured, step-by-step reasoning process. PTS mitigates the shortcomings of fast perception through two key components: (1) Perception Elaboration, which encourages models to produce token-rich descriptions of perceptual outcomes by using symbolic tokens to represent abstract visual attributes (*e.g.,* distance), thereby providing more grounded and interpretable representations than raw numerical values. (2) Perception Decomposition, which trains models to decompose complex perception tasks into simpler sub-problems.

We instantiate PTS through a two-stage training pipeline. In the first stage, a supervised cold-start phase teaches the model the structured perception pattern. In the second stage, we apply reinforcement learning with GRPO (Shao et al., 2024) to further optimize the model, enabling it to explore and refine intermediate perceptual steps. Our experiments demonstrate that PTS enables inference-time scaling to benefit not only reasoning but also perception. Specifically, PTS yields substantial gains on DisTANCE, improving high-precision performance from 8.0% to 64.7%, while also generalizing to out-of-domain visual estimation and broader multimodal tasks. Moreover, although PTS data are entirely synthetic, their integration with math reasoning data consistently improves both reasoning and real-world perception benchmarks. Further analysis reveals that PTS introduces more perception-related tokens and increases the model's attention to image tokens, explaining how it enhances perceptual competence. These findings demonstrate that explicitly modeling perception as a structured process not only enhances interpretability, but also unlocks significant perceptual improvements from inference-time scaling, paving the way toward more perception-aware LVLMs.

## 2 DISTANCE: PROBING PERCEPTION OF LVLMS WITH VISUAL ESTIMATION

To examine whether inference-time scaling can enhance visual perception, we introduce **DisTANCE** (**Dis**tance reasoning **T**ask with **A**nalytical **N**umeric and **C**omparative **E**stimation).

### 2.1 OVERVIEW

DisTANCE focuses on a conceptually simple yet diagnostically effective task: *visual estimation*, which requires models to estimate geometric attributes such as length, perimeter, and area in synthetic

Table 1: Evaluation results of proprietary, open-source and reasoning-oriented LVLMs on DisTANCE.

| Model | Length | | Perimeter | | Area | | Average | |
|---|---|---|---|---|---|---|---|---|
| | $RA_{0.1}$ | $RA_{avg}$ | $RA_{0.1}$ | $RA_{avg}$ | $RA_{0.1}$ | $RA_{avg}$ | $RA_{0.1}$ | $RA_{avg}$ |
| *Proprietary Models* | | | | | | | | |
| Claude-3.5-Sonnet (Claude, 2024) | 25.0 | 55.0 | 23.0 | 55.0 | 9.0 | 15.6 | 19.0 | 41.9 |
| Gemini-2.5-Flash (Google, 2025) | 36.0 | 64.2 | 20.0 | 57.4 | 14.0 | 34.0 | 23.3 | 51.9 |
| Gemini-2.5-Pro (Google, 2025) | 36.0 | 67.8 | 32.0 | 69.2 | **21.0** | **50.4** | 29.7 | 62.5 |
| GPT-4o (OpenAI, 2024) | 22.0 | 47.4 | 13.0 | 37.6 | 11.0 | 22.0 | 15.3 | 35.7 |
| GPT-4.1 (OpenAI, 2025a) | 32.0 | 63.2 | 19.0 | 55.4 | 10.0 | 33.2 | 20.3 | 50.6 |
| GPT-5 (OpenAI, 2025b) | **41.0** | **72.6** | **37.0** | **72.8** | 17.0 | 46.8 | **31.7** | **64.1** |
| *Open-source Models* | | | | | | | | |
| Qwen2.5-VL-3B (Bai et al., 2025) | 11.0 | 26.8 | 6.0 | 18.6 | 2.0 | 13.0 | 6.3 | 19.5 |
| Qwen2.5-VL-7B (Bai et al., 2025) | 11.0 | 25.6 | 11.0 | 27.8 | 2.0 | 11.2 | 8.0 | 21.5 |
| Qwen2.5-VL-32B (Bai et al., 2025) | 14.0 | 38.6 | 11.0 | 26.8 | 5.0 | 16.4 | 10.0 | 27.3 |
| Qwen2.5-VL-72B (Bai et al., 2025) | 16.0 | 42.0 | 8.0 | 34.8 | 6.0 | 13.4 | 10.0 | 30.1 |
| InternVL3-8B (Zhu et al., 2025) | 11.0 | 29.2 | 8.0 | 21.2 | 5.0 | 11.8 | 8.0 | 20.7 |
| InternVL3-14B (Zhu et al., 2025) | 16.0 | 42.0 | 10.0 | 37.0 | 9.0 | 14.4 | 11.7 | **31.1** |
| InternVL3-38B (Zhu et al., 2025) | **19.0** | **45.2** | 10.0 | 30.6 | 4.0 | 14.8 | 11.0 | 29.9 |
| InternVL3-78B (Zhu et al., 2025) | 11.0 | 35.6 | **17.0** | **38.8** | 10.0 | 17.2 | 12.7 | 30.5 |
| *Open-source Reasoning Models* | | | | | | | | |
| R-4B (Yang et al., 2025a) | 11.0 | 28.2 | 9.0 | 27.4 | 8.0 | 19.2 | 9.3 | 24.9 |
| R1-OneVision-7B (Yang et al., 2025b) | 9.0 | 23.4 | 12.0 | 26.0 | 9.0 | 14.0 | 10.0 | 21.1 |
| Vision-R1-7B (Huang et al., 2025) | 9.0 | 27.0 | 8.0 | 25.2 | 6.0 | 16.8 | 7.7 | 22.7 |
| MM-Eureka-7B (Meng et al., 2025) | 6.0 | 20.6 | 10.0 | 27.6 | 3.0 | 10.2 | 6.3 | 19.5 |
| VLAA-Thinker-7B (Chen et al., 2025a) | 10.0 | 31.8 | 9.0 | 31.8 | 7.0 | 15.0 | 8.7 | 26.2 |
| NoisyRollout-7B (Liu et al., 2025b) | **12.0** | **34.8** | 9.0 | 30.4 | 6.0 | 15.8 | 9.0 | **27.0** |
| VL-ReThinker-7B (Wang et al., 2025a) | 11.0 | 28.0 | **15.0** | **37.8** | 4.0 | 14.0 | 10.0 | 26.6 |
| ReVPT-7B (Zhou et al., 2025b) | **12.0** | 25.6 | 13.0 | 29.2 | 9.0 | 15.4 | **11.3** | 23.4 |

images. Since answers rely solely on visual perception, DisTANCE serves as a focused benchmark to test whether current reasoning paradigms also enhance perception.

**Task Formulation.** As illustrated in Figure 1, DisTANCE contains tasks that require precise estimation of geometric measurements in synthetic images. Each image includes several geometric objects (*e.g.,* circles, triangles, rectangles) with varied colors, sizes, and orientations. The model is prompted to predict quantitative relations between spatial properties in a regression manner (*e.g.,* "What is the height of the image if the radius of the red circle is 1 unit?"), requiring it to perceive and compare distances with numerical precision. To remove confounding effects from image resolution, all tasks are formulated as relative estimations.

**Benchmark Details.** DisTANCE consists of 300 image-question pairs, with each sub-task including 100 examples. All images are automatically synthesized using Python with dimensions ranging from 600 to 1200 pixels, and each image contains 3 to 7 colored shapes. The shapes include circles, triangles, and rectangles filled with distinct colors. Each sub-task has multiple different question templates. More cases are presented in the Appendix K.

## 2.2 EVALUATION SETUP

**Metrics.** Following existing work that evaluates LVLMs using numerical answers (Yang et al., 2024), we adopt Relative Accuracy (RA) as the evaluation metric. A prediction $\hat{y}$ is considered correct under threshold $\theta$ if the relative error $|\hat{y} - y|/y < \theta$. We report $RA_{avg}$ under various thresholds as:

$$\text{RA}_{\text{avg}}(\hat{y}) = \frac{1}{|C|} \sum_{\theta \in C} \mathbb{K}\left(\frac{|\hat{y} - y|}{y} < \theta\right), \tag{1}$$

where $C = \{0.5, 0.4, 0.3, 0.2, 0.1\}$. We also report $RA_{0.1}$ for high-precision performance.

Table 2: Comparison on the average response length and perception ratio (PR) of base and inference-time scaled (ITS) LVLMs.

| Model | ITS | Avg Len. | PR |
|-------|-----|----------|-----|
| Qwen2.5-VL-7B | ✗ | 199 | 17.6 |
| R1-OneVision-7B | ✓ | 322 | 12.4 |
| MM-Eureka-7B | ✓ | 259 | 13.5 |

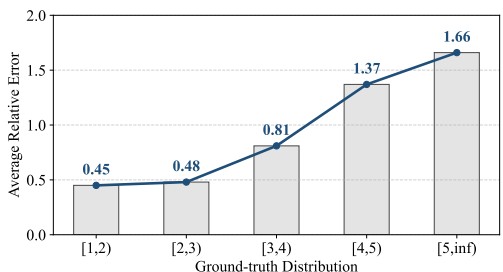

Figure 2: Relative error trend across labels.

**Evaluated Models.** We evaluate a range of proprietary and open-source LVLMs, including GPT-4o (OpenAI, 2024), GPT-4.1 (OpenAI, 2025a), GPT-5 (OpenAI, 2025b), Gemini-2.5 (Google, 2025), Claude-3.5 (Claude, 2024), Qwen2.5-VL (Bai et al., 2025), and InternVL3 (Zhu et al., 2025), as well as reasoning-oriented LVLMs such as R1-OneVision (Yang et al., 2025b), Vision-R1 (Huang et al., 2025), MM-Eureka (Meng et al., 2025), VLAA-Thinker (Chen et al., 2025a), NoisyRollout (Liu et al., 2025b), VisualThinker (Zhou et al., 2025a), R-4B (Yang et al., 2025a), VL-ReThinker (Wang et al., 2025a) and ReVPT (Zhou et al., 2025b).

## 2.3 EVALUATION RESULTS

As shown in Table 1, most LVLMs perform poorly on DisTANCE: open-source models rarely exceed an $RA_{avg}$ of 35%. Advanced proprietary LVLMs, including Gemini-2.5-Pro and GPT-5, are capable of utilizing tools during inference, resulting in significantly better performance. Area estimation is particularly difficult, with most models below 20%. Scaling model size has limited effect, and reasoning-enhanced LVLMs show only marginal gains over their bases. For example, Vision-R1 (22.7%) and R1-OneVision (21.1%) perform on par with Qwen2.5-VL-7B (21.5%). These results suggest that current inference-time scaling paradigms, although effective at generating longer reasoning chains, do not significantly enhance performance on perception-centric tasks. To further investigate this issue, we conduct additional analysis and identify two key findings.

**LVLMs Prefer Fast Perception.** By observing generated cases, we notice that LVLMs often summarize perceptual information using short, fixed patterns (*e.g.,* "the radius of the circle is 2.5 units"), without further elaboration. We refer to this behavior as *Fast Perception*, a tendency to express perception results in minimal tokens. To quantify this, we analyze the ratio of perception-related tokens (formally defined in Appendix B) and the average length of model outputs. As shown in Table 5, reasoning-oriented models produce longer answers but allocate only a small fraction to perception. This highlights a limitation of current inference-time scaling: it scales reasoning but not perception, motivating paradigms that explicitly scale both.

**LVLMs Fail to Decompose Complex Perception.** We further investigate a key failure mode of LVLMs in complex perception tasks. Specifically, we group length-estimation questions by ground-truth distance and measuring Qwen2.5-VL-7B's relative error. As shown in Figure 2, the model's relative error increases consistently with target length. This trend suggests that while inference-time scaling teaches models to reason step by step, such decomposition ability does not transfer to the perception stage. Even for complex perception cases, models still prefer fast perception, directly producing an answer without any intermediate process or decomposing the task.

## 3 METHODOLOGY

Previous empirical results reveal a key limitation: The prevailing fast perception paradigm limits inference-time scaling from improving perception abilities of reasoning-enhanced LVLMs. To address this, we propose a new perception paradigm, **Perception-Time Scaling (PTS)**, which aims to structurally enhance perception in a way that is compatible with inference-time scaling techniques. In this section, we define the PTS framework and describe how we apply it to the visual estimation tasks from the proposed DisTANCE. The overview of PTS is illustrated in Figure 3.

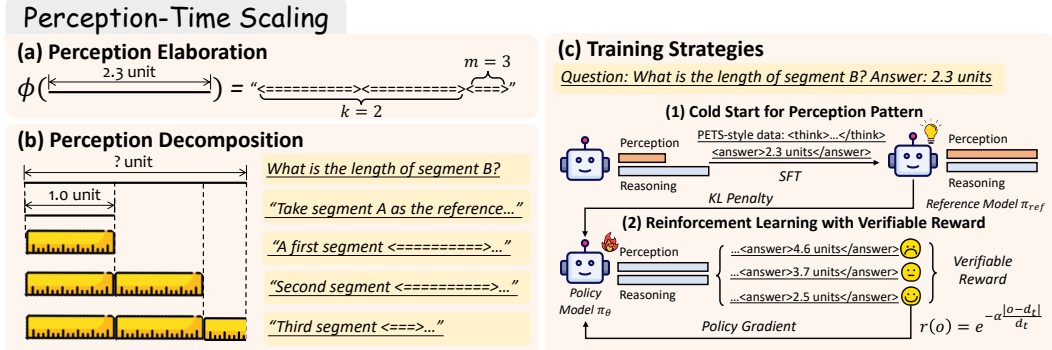

Figure 3: Overview of the proposed PTS paradigm. In the context of visual estimation tasks, PTS first employs symbolic tokens to intuitively represent distance information, followed by a step-by-step decomposition of the estimation process. During training, LVLMs are initially taught the PTS pattern via SFT, then further optimized using GRPO to enhance perception accuracy.

### 3.1 PERCEPTION-TIME SCALING

To overcome the limitations of the Fast Perception paradigm, PTS introduces two complementary strategies. **(1) Perception Elaboration.** PTS encourages models to produce token-rich descriptions of perceptual outcomes. Specifically, abstract visual attributes such as distance are represented using symbolic tokens, which provide more grounded and interpretable descriptions than raw numerical values. **(2) Perception Decomposition.** PTS teaches models to decompose complex perception tasks into simpler and more tractable sub-problems. This stepwise approach enables more accurate visual understanding by mimicking how humans often solve challenging perception tasks through progressive comparison. We next describe the detailed implementation of PTS in the context of visual quantitative estimation tasks.

**Perception Elaboration.** Given a target distance for estimation $d_t$, we define a symbolic encoding function $\phi$, which maps $d_t$ into a discrete token sequence $\phi(d_t)$. $\phi(d_t)$ consists of several symbolic token groups rendered as "<==========>", each representing distance of 1.0 unit. The angle brackets < and > are delimiters and each equal sign = represents a fixed length $\delta = 0.1$ unit. Given $d_t \in \mathbb{R}_{\geq 0}$, we decompose it as: $k = \left\lfloor \frac{d_t}{1.0} \right\rfloor, \quad r = d_t - k$, where $k$ is the number of full symbolic token group and $r \in [0, 1.0)$ is the residual length. The encoded sequence is represented as:

$$\phi(d_t) = \underbrace{\text{<==========>} \cdots \text{<==========>}}_{k \text{ times}} \| \phi_{\text{res}}(r). \tag{2}$$

$\phi_{\text{res}}(r)$ is the symbolization results for $r$, which is defined as:

$$\phi_{\text{res}}(r) = \text{<} \underbrace{= \cdots =}_{m \text{ times}} \text{>} \tag{3}$$

where $m = \left\lfloor \frac{r}{\delta} \right\rfloor$. By doing so, PTS increases the presence of perception tokens in model outputs and creates stronger grounding between vision and language. Rather than a single numerical token, the model must now construct the perception output step by step, allowing more intuitive and fine-grained representation of distance information.

**Perception Decomposition.** Previous experiments show that even when the task complexity increases, models continue to rely on one-shot perception. To mitigate this limitation, PTS introduces a decomposition strategy that breaks down complex perception tasks into step-by-step local subgoals. Specifically, given the target distance $d_t$, we first select a reference segment $d_r$ from the image, whose length is defined as 1.0 unit. Rather than directly predicting the value of $d_t$, we ask the model to perform a step-by-step accumulation of $d_r$ until fully covering the $d_t$, which can be formulated as:

$$\begin{aligned} \text{Initialize:} \quad & L = 0, \quad k = 0; \\ \text{While } L + d_r \leq d_t : \quad & L \leftarrow L + d_r, \quad k \leftarrow k + 1. \end{aligned} \tag{4}$$

This process lets models simulate the process of tiling $d_r$ along $d_t$ and incrementally accumulating distance until the target is fully covered. Each step in this process is relatively simple and localized, making the overall task more tractable. This approach also mirrors how humans handle complex measuring tasks with a ruler. Moreover, by transforming perception into a step-by-step process, this design opens up greater opportunities for inference-time scaling, allowing the model to optimize each perception step and ultimately achieve more accurate visual understanding.

## 3.2 Cold Start for Perception Pattern

We begin by defining the reasoning process under the PTS paradigm. Each reasoning chain follows five stages: (1) **Review**: A brief description of the task; (2) **Hint**: Definition for the symbolic encoding and decomposition strategy; (3)**Reference**: Selection of a specific segment in the image to serve as a reference; (4) **Estimation**: Visual comparison of other segments to the reference to estimate their lengths; (5) **Calculation**: Application of appropriate formulas to compute the final result.

To teach LVLMs this structured perception reasoning pattern, we construct a training dataset in PTS style. Following the data collection pipeline of DisTANCE, we synthesize images and corresponding questions for length, perimeter, and area estimation tasks. Each reasoning chain is formatted in line with (Guo et al., 2025), where the reasoning steps are enclosed within `<think></think>` tags and the final answer within `<answer></answer>` tags. The dataset is synthesized through a two-step procedure. We first manually collect a small set of reasoning examples for each task, serving as few-shot demonstrations. Next, we feed the distance information within images together with the demonstrations, to GPT-4o to generate instructions. This process yields 2,000 examples per task, resulting in a total of 6,000 PTS-style reasoning chains.

Importantly, all training images are freshly synthesized with random seeds to avoid overlap with evaluation benchmarks. The resulting dataset is then used for supervised fine-tuning (SFT), enabling the model to acquire an initial understanding of the PTS perception pattern. A full example of the reasoning process and the GPT-4o prompting template is provided in Appendix I.

## 3.3 Reinforcement Learning with Verifiable Reward

To further enhance perception quality under the PTS paradigm, we continue optimizing the model after the cold start stage using GRPO (Shao et al., 2024), a reinforcement learning method with verifiable reward that supports inference-time scaling and improves reasoning capabilities.

**Preliminary of GRPO.** GRPO is a reinforcement learning algorithm, which adopts a verifiable reward function instead of a reward model, and calculates the relative advantages within a group of completions for a given sample. Specifically, GRPO first samples a group of answers $\{o_1, o_2, \cdots, o_N\}$ from the old policy model $\pi_{\theta_{\text{old}}}$ for each question $q$. The policy model $\pi_\theta$ is then optimized by:

$$\mathcal{J}(\theta) = \frac{1}{N} \sum_{i=1}^{N} \left( \min \left( \frac{\pi_\theta(o_i|q)}{\pi_{\theta_{\text{old}}}(o_i|q)} A_i, \text{clip} \left( \frac{\pi_\theta(o_i|q)}{\pi_{\theta_{\text{old}}}(o_i|q)}, 1-\epsilon, 1+\epsilon \right) A_i \right) - \beta \mathbb{D}_{KL}(\pi_\theta||\pi_{\text{ref}}) \right), \quad (5)$$

where $\epsilon$ and $\beta$ are hyperparameters and $A_i$ is the advantage of $o_i$ in the group. $A_i$ is calculated by:

$$A_i = \frac{r_i - \text{mean}(\{ r_1, r_2, \cdots, r_N \})}{\text{std}(\{ r_1, r_2, \cdots, r_N \})}, \quad (6)$$

where $r_i$ is the reward given by a rule-based reward function. $r_i$ is typically set as $r = \lambda r_{\text{acc}} + (1 - \lambda) r_{\text{format}}$. Here, $r_{\text{acc}}$ is the accuracy reward which is set to 1 if the model's prediction is correct, $r_{\text{format}}$ is the format reward which is set to 1 if the model follows the expected response format, and $\lambda$ is the hyperparameter controls the ratio between two rewards. In PTS, the perception process is explicitly embedded in the reasoning chain, allowing reward signals to be applied to intermediate perception steps, enabling the model to gradually improve its perception accuracy through step-by-step refinement. We further adapt GRPO to our task with the following modifications:

**Customized Reward for Regression.** Since tasks in DisTANCE involve continuous regression, binary accuracy rewards are insufficient to capture how close a prediction is to the ground-truth.

Table 3: Effect of different training data paradigms and strategies on DisTANCE. *Direct*, *CoT*, *PTS* represent the training data in direct answer, Chain-of-Thought and PTS format, respectively. SFT and RL represent whether to adopt supervised fine-tuning or GRPO for training.

| Model | SFT | RL | Length | | Perimeter | | Area | | Average | |
|---|---|---|---|---|---|---|---|---|---|---|
| | | | $RA_{0.1}$ | $RA_{avg}$ | $RA_{0.1}$ | $RA_{avg}$ | $RA_{0.1}$ | $RA_{avg}$ | $RA_{0.1}$ | $RA_{avg}$ |
| SpaceThinker-3B | - | - | 7.0 | 39.0 | 8.0 | 38.0 | 6.0 | 26.0 | 7.0 | 34.3 |
| Spatial-R1-7B | - | - | 15.0 | 51.0 | 7.0 | 39.0 | 6.0 | 32.0 | 9.3 | 40.7 |
| GPT-4o + DetToolChain | - | - | 24.0 | 53.8 | 16.0 | 47.8 | 6.0 | 24.4 | 15.3 | 42.0 |
| GPT-4o + Sketchpad | - | - | 27.0 | 36.6 | 20.0 | 34.5 | 10.0 | 19.2 | 19.0 | 30.1 |
| Qwen2.5-VL-3B | - | - | 11.0 | 26.8 | 6.0 | 18.6 | 2.0 | 13.0 | 6.3 | 19.5 |
| + *Direct* | ✓ | ✗ | 16.0 | 44.3 | 24.0 | 57.8 | 14.0 | 30.2 | 18.0 | 44.1 |
| + *CoT* | ✓ | ✗ | 15.0 | 48.0 | 28.0 | 61.8 | 13.0 | 34.2 | 18.7 | 48.0 |
| + *PTS* | ✓ | ✗ | 14.0 | 36.0 | 20.0 | 46.8 | 9.0 | 24.6 | 14.3 | 35.8 |
| + *Direct* | ✗ | ✓ | **36.0** | 65.6 | 44.0 | 76.8 | 13.0 | 32.8 | 35.8 | 47.5 |
| + *CoT* | ✓ | ✓ | 29.0 | 62.7 | 46.0 | 79.2 | 16.0 | 38.7 | 27.0 | 60.2 |
| + *PTS* | ✓ | ✓ | 31.0 | **70.6** | 54.0 | 86.2 | 28.0 | 63.6 | 37.7 | 73.5 |
| Qwen2.5-VL-7B | - | - | 11.0 | 25.6 | 11.0 | 27.8 | 2.0 | 11.2 | 8.0 | 21.5 |
| + *0-shot* | ✗ | ✗ | 3.0 | 12.0 | 7.0 | 19.8 | 5.0 | 13.4 | 5.0 | 15.1 |
| + *1-shot* | ✗ | ✗ | 5.0 | 17.8 | 8.0 | 23.2 | 3.0 | 16.2 | 5.3 | 19.1 |
| + *4-shot* | ✗ | ✗ | 12.0 | 25.8 | 14.0 | 26.0 | 5.0 | 13.6 | 10.3 | 21.8 |
| + *8-shot* | ✗ | ✗ | 4.0 | 16.6 | 3.0 | 15.8 | 6.0 | 15.6 | 4.3 | 16.0 |
| + *Direct* | ✓ | ✗ | 20.0 | 49.4 | 28.0 | 63.0 | 16.0 | 33.4 | 21.3 | 48.6 |
| + *CoT* | ✓ | ✗ | 17.0 | 40.8 | 10.0 | 52.2 | 14.0 | 29.2 | 13.7 | 40.7 |
| + *PTS* | ✓ | ✗ | 12.0 | 44.2 | 25.0 | 62.0 | 12.0 | 30.2 | 16.3 | 45.4 |
| + *Direct* | ✗ | ✓ | 46.0 | 77.4 | 51.0 | 82.4 | 25.0 | 52.4 | 40.7 | 70.7 |
| + *CoT* | ✓ | ✓ | 38.0 | 74.6 | 48.0 | 80.8 | 27.0 | 58.6 | 37.7 | 71.3 |
| + *PTS* | ✓ | ✓ | **70.0** | **92.2** | **74.0** | **94.6** | **50.0** | **78.0** | **64.7** | **88.3** |

Moreover, we expect the reward to explicitly encourage high-precision estimation. To address this, we design a continuous reward function based on relative error in exponential form:

$$r(o) = e^{-\alpha \frac{|o-d_t|}{d_t}}, \tag{7}$$

where $o$ is the model's prediction, $d_t$ is the ground-truth value, and $\alpha$ is a hyperparameter controlling the sensitivity of the reward. This formulation grants higher rewards to predictions with smaller relative errors. The exponential structure makes the reward particularly sensitive to small errors, thus incentivizing fine-grained accuracy. We provide the visualization of the reward function and ablation studies on the choice of $\alpha$ in the Appendix C.

**Label Normalization.** We further observe that the same relative error threshold can correspond to very different absolute error tolerances, depending on the scale of the label value. For example, a 10% relative error threshold on a distance of 0.02 permits an absolute error of only $\pm 0.002$, while the same threshold on a value of 50 allows a $\pm 5$ deviation. Consequently, the reward function could assign the same reward to estimations with significantly different levels of precision, which may confuse the model at the beginning of training. To address this, we propose to first train the model on normalized samples with target values less than 1, and later introduce data with random distributions.

## 4 EXPERIMENT

### 4.1 EXPERIMENTAL SETTINGS

**Baselines.** We compare the proposed PTS against three categories of baselines: (1) **Prompting strategy.** We reformulate the data collection process of PTS into a system prompt for the model and provide different numbers of PTS reasoning chains as demonstrations (*k-shot*). (2) **Different data patterns.** Using the same image source, we construct two alternative instruction datasets: one that only includes the final numerical answer (*Direct*), and another that contains conventional chain-of-thought annotations synthesized by GPT-4o (*CoT*). These datasets are used to train the base LVLMs

Table 4: Experimental results on out-of-domain distance reasoning tasks. LHC is the LineComparison subset from Geoperception. Height is the Height subset from LEGO-Puzzles.

| Model | SFT | RL | Length$_{ood}$ | | Perimeter$_{ood}$ | | Geo$_{LHC}$ | LEGO$_{Height}$ |
|---|---|---|---|---|---|---|---|---|
| | | | RA$_{0.1}$ | RA$_{avg}$ | RA$_{0.1}$ | RA$_{avg}$ | Accuracy | Accuracy |
| Qwen2.5-VL-3B | - | - | 6.0 | 12.2 | 10.0 | 23.8 | 38.9 | 28.0 |
| + *CoT* | ✓ | ✗ | 8.0 | 29.8 | 16.0 | 45.0 | 64.5 | 27.0 |
| + *PTS* | ✓ | ✗ | 10.0 | 32.6 | 7.0 | 31.8 | 68.4 | 22.0 |
| + *Direct* | ✗ | ✓ | 12.0 | 38.9 | 23.0 | 52.4 | 70.4 | 23.0 |
| + *CoT* | ✓ | ✓ | 12.0 | 37.6 | 24.0 | 53.4 | 71.1 | 30.0 |
| + *PTS* | ✓ | ✓ | **16.0** | **41.4** | **26.0** | **55.0** | **73.5** | **44.0** |
| Qwen2.5-VL-7B | - | - | 7.0 | 22.2 | 12.0 | 23.6 | 59.8 | 30.0 |
| + *CoT* | ✓ | ✗ | 14.0 | 33.8 | 21.0 | 52.0 | 65.6 | 29.0 |
| + *PTS* | ✓ | ✗ | 15.0 | 36.6 | 26.0 | 56.0 | 67.1 | 28.0 |
| + *Direct* | ✗ | ✓ | 19.0 | 40.2 | 30.0 | 63.4 | 69.4 | 30.0 |
| + *CoT* | ✓ | ✓ | 18.0 | 39.6 | 20.0 | 55.6 | 72.2 | 30.0 |
| + *PTS* | ✓ | ✓ | **20.0** | **43.2** | **36.0** | **71.4** | **78.7** | **33.0** |

with both SFT and GRPO. (3) **Spatial-aware LVLMs.** Models explicitly designed to estimate spatial information such as object sizes in real-world scenarios, including Spatial-R1 (Ouyang, 2025) and SpaceThinker (remyxai, 2025). (4) **Tool-augmented LVLMs.** Approaches that integrate external tools (*e.g.,* visual experts or Python code) to provide auxiliary visual information, such as Visual Sketchpad (Hu et al., 2024) and DetToolChain (Wu et al., 2024b). Implementation details are presented in Appendix J. For backbone models, we select the state-of-the-art LVLMs Qwen2.5-VL-3B and Qwen2.5-VL-7B, and implement our method on them.

**Evaluated Benchmarks.** We evaluate model's performance on DisTANCE and other out-of-domain benchmarks: (1) **DisTANCE$_{ood}$**: we replace the images in DisTANCE shapes not included in the collected training data (*e.g.,* trapezoids and pentagons); (2) **Geoperception** (Zhang et al., 2024): a fine-grained perception benchmark. We select the LHC subset which requires the model to compare the length of two segments in geometric images; (3) **LEGO-Puzzles** (Tang et al., 2025a): a spatial understanding benchmark with LEGO-based tasks, where we select the Height subset that asks the model to distinguish the relative heights of 3D LEGO objects.

## 4.2 MAIN RESULTS

We present the experimental results in Table 3 and 4, from which we conclude:

**PTS Enables High-Precision Visual Estimation.** Across both the 3B and 7B versions of Qwen2.5-VL, models trained with PTS data consistently outperform all baselines. For instance, when combined with SFT and RL, PTS significantly boosts the average RA$_{avg}$ of Qwen2.5-VL-7B from 21.5% to 88.3%. Notably, under the strict threshold (RA$_{0.1}$), the model achieves 70.0%, 74.0%, and 50.0% accuracy on the length, perimeter, and area subsets, respectively, demonstrating strong high-precision estimation capabilities. It is also worth noting that PTS strongly outperforms other baselines, despite not using any spatial-related data or external tools. Instead of relying on external modules for spatial perception, PTS internalizes the perception process into the reasoning chain through symbolic tokens, allowing it to be optimized end-to-end during training, which leads to more accurate and robust estimation. Additional qualitative examples are provided in the Appendix K and L.

**PTS Generalizes to Out-of-Domain Estimation Tasks.** Table 4 demonstrates that PTS exhibits strong generalization capabilities across various estimation tasks beyond the training domain. Specifically, PTS improves the model's estimation performance on geometric shapes not seen during training. It also enhances performance in entirely unseen scenarios. For complex geometric shapes in the Geoperception, PTS boosts the model's accuracy by approximately 20%. Furthermore, in 3D scenarios from the LEGO-puzzles, PTS improves the model's accuracy in judging the height of objects, highlighting the potential of our method in real-world scenarios.

Table 5: Performance comparison of Qwen2.5-VL variants on general multimodal benchmarks.

| Model | MathVision | MMBench | MMVet | HalluBench | CV-Bench | BLINK | | | |
| --- | --- | --- | --- | --- | --- | --- | --- | --- | --- |
| | | | | | | Jigsaw | Multi | Local | Full |
| Qwen2.5-VL | 25.3 | 83.28 | 66.93 | 51.2 | 73.36 | 66.7 | 46.6 | 48.4 | 55.0 |
| + Geo3K | 26.8 | 83.82 | 67.11 | 52.2 | 74.30 | 66.2 | 52.6 | 52.5 | 54.3 |
| + PTS, Geo3K | **27.2** | **85.68** | **67.39** | **53.0** | **75.68** | **68.9** | **53.4** | **53.3** | **55.9** |

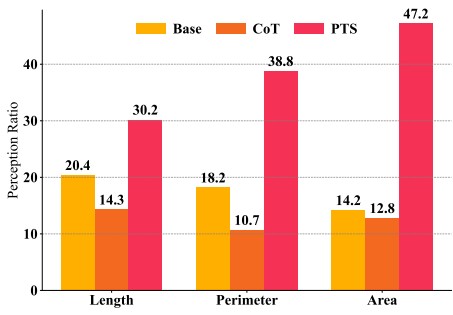

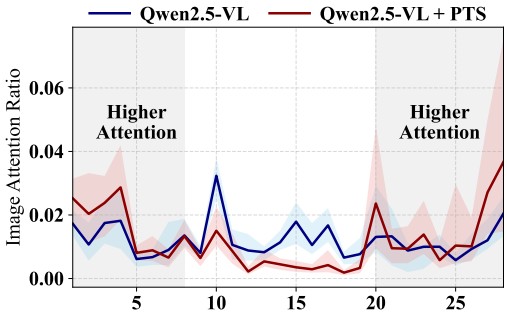

(a) Perception Ratios of Qwen2.5-VL-7B Variants     (b) Image attention ratio across transformer layers

Figure 4: Influence of PTS on model's perception pattern.

## 4.3 INFLUENCE ON PERCEPTION CAPABILITIES

To further investigate the influence of PTS on models' perception capabilities, we validate its effectiveness on more diverse multimodal datasets from multiple domains. We also confirm that PTS data indeed enhances models' perceptual ability by examining both the explicit perception token ratio and the intrinsic image attention distribution. Ablation studies on symbolic token designs, granularity, training strategy and data efficiency analysis are presented in Appendix E, F and G.

**PTS Improves Both General and Perception-Centric Multimodal Tasks.** PTS further enhances the general multimodal capabilities of LVLMs when combined with math reasoning data from Geometry3K (Lu et al., 2021). We apply GRPO to a cold-started Qwen2.5-VL-7B model and evaluate on MathVision (Wang et al., 2024) for math reasoning, MMBench (Liu et al., 2024) and MMVet (Yu et al., 2024) for general multimodal VQA, HallusionBench (Guan et al., 2024) for hallucination, and CV-Bench (Tong et al., 2024a) and BLINK (Fu et al., 2024) for perception-centric tasks. As shown in Table 5, models trained only on math reasoning data improve performance on math-related tasks but yield limited or even negative gains on perception-heavy benchmarks. By contrast, incorporating PTS data not only strengthens reasoning ability but also brings consistent improvements across diverse domains—most notably on CV-Bench and BLINK, where fine-grained perception is critical. Remarkably, despite consisting solely of synthetic images, PTS still transfers effectively to real-world perception tasks, highlighting its strong generalization capacity.

**PTS Facilitates Effective Perception Refinement.** While both PTS and CoT yield similar performance after SFT, their differences become evident during reinforcement learning. PTS sequences explicitly embed perception steps into the reasoning chain, offering a richer structure for optimization. During GRPO training, models can explore and refine these intermediate perception results, leading to substantial improvements in estimation accuracy. In contrast, CoT sequences contain minimal perceptual content, limiting the scope for refinement and resulting in marginal gains. As illustrated in Figure 4a, PTS significantly increases the proportion of perception-related tokens, offering RL more opportunities to enhance perceptual accuracy.

**PTS Strengthens Image-Focused Attention.** We further investigate how PTS training reshapes the model's attention distribution over image tokens. Specifically, we compare the attention ratio of the first perception token in vanilla Qwen2.5-VL with that of the symbolic tokens introduced by PTS-trained models. The data are collected from the generation results on both DisTANCE and OOD tasks. As shown in Figure 4b, PTS-trained models consistently assign higher attention to image regions, especially in the early and final layers. This pattern suggests that PTS enhances low-level

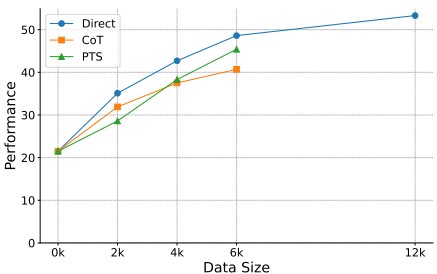

Figure 5: Impact of scaling training data.

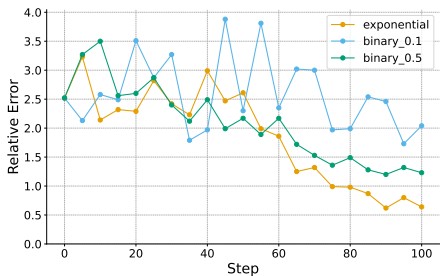

Figure 6: Error curve with different rewards.

grounding at the input stage while also reinforcing image-conditioned reasoning in the later decoding process. Together, these results indicate that PTS encourages the model to more effectively integrate visual evidence throughout the perception–reasoning pipeline.

### 4.4 ADDITIONAL ANALYSIS

**Impact of Training Data Scaling.**    We study the impact of scaling the training data on the model's performance. We conduct SFT on different amount of Direct (2k-12k), CoT (2k-6k), and PTS (2k-6k) data. The performance curve is presented in Figure 5. Initially, scaling the data results in noticeable improvements, but as the dataset size increases, the performance gains gradually plateau. This indicates that simply scaling the data is insufficient for visual estimation tasks.

**Ablation on Reward Function.**    In addition to the proposed continuous reward, we also test a discrete binary reward defined as $r(o) = 1$ if $|o - d_t|/d_t < \tau$, and $r(o) = 0$ otherwise, where $\tau$ is the threshold. We evaluate two values of $\tau$: $0.1$ for a stricter threshold and $0.5$ for a more relaxed one. Based on Qwen2.5-VL-7B, we conduct 100 steps of GRPO with all three reward functions and compare their effects on relative error changes in the validation set, as shown in Figure 6. The binary reward with $\tau = 0.1$ performs the worst, exhibiting significant oscillations and minimal error reduction. This is likely due to the strict threshold, which prevents most training data from providing useful feedback. Increasing $\tau$ to 0.5 results in smoother error reduction but slower convergence, as the binary reward does not differentiate between responses of varying precision. In contrast, the continuous exponential reward converges the fastest, achieving a final relative error that is less than 50% of that of the binary reward with $\tau = 0.5$.

## 5 CONCLUSION

In this paper, we investigated how inference-time scaling techniques affect the perception capabilities of LVLMs. To this end, we introduced DisTANCE, a lightweight yet diagnostic benchmark designed to evaluate LVLMs with visual estimation tasks. Our evaluation revealed that, under the prevailing fast perception paradigm in LVLMs, inference-time scaling offers limited perception improvements. To address this limitation, we proposed PTS, a novel paradigm that makes the perception process more structured and explicit via perception elaboration and decomposition. After two-stage inference-time scaling with SFT and RL, PTS enables substantial perception improvements. Empirical results showed that PTS consistently enhances performance on both in-domain and out-of-domain visual estimation tasks. Further analysis confirmed the generality of the PTS in general multimodal tasks, suggesting a promising direction for future research on inference-time-enhanced perception in LVLMs.

## 6 ACKNOWLEDGEMENTS

This paper was partially supported by the National Natural Science Foundation of China No. 92470205 and Beijing Major Science and Technology Project under Contract No. Z251100008425002. Xin Zhao and Minghui Qiu are the corresponding authors.

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

## A    USE OF LARGE LANGUAGE MODELS

In this paper, we use large language models only for minor editing tasks in writing, *i.e.,* improving readability and grammar, and for occasional debugging hints in coding. The core contributions, including the proposed benchmark, algorithm design, experimental setup, analysis, and conclusions, were fully developed and verified by the authors.

## B    DEFINITION OF PERCEPTION RATIO

We define *perception tokens* as those directly describing visual content in the input image, including:

1. **Explicit geometric attributes**, such as shape mentions (*e.g.,* "circle," "line," "triangle");

2. **Numerical values tied to measurements**, such as "2.5 units," "length of 3 cm,";

3. **Symbolic markers** introduced by our framework (*e.g.,* "<====>" tokens).

We implement this definition using a keyword-based filter over tokenized model outputs. For each model, we randomly sample 50 generated outputs for calculation.

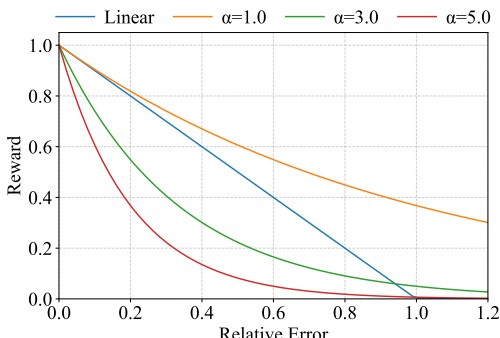

Figure 7: Curve of linear reward and exponential reward with different $\alpha$.

Table 6: Performance under varying $\alpha$

| Parameter | DisTANCE | |
| --- | --- | --- |
| | $RA_{0.1}$ | $RA_{avg}$ |
| $\alpha = 1.0$ | 53.0 | 75.2 |
| $\alpha = 3.0$ | **64.7** | **88.3** |
| $\alpha = 5.0$ | 36.0 | 72.4 |

## C    ANALYSIS ON REWARD FACTOR $\alpha$

We illustrate the curves of the linear reward and the exponential reward under different values of $\alpha$ in Figure 7. As shown, when $\alpha$ is appropriately set (*e.g.,* $\alpha = 3$ as used in our experiments), the exponential reward remains low and varies smoothly for large relative errors (*e.g.,* >0.6), while exhibiting a steep increase for small relative errors (*e.g.,* <0.2). Compared to the linear reward, this design encourages the model to make high-precision estimations. We further conduct an ablation study on the choice of $\alpha$ by comparing model performance under $\alpha = 1$, 3, and 5. As shown in Table 6, both smaller and larger values of $\alpha$ lead to degraded performance compared to our chosen setting. We suggest that when $\alpha$ is too small, the model receives overly high rewards even at the beginning of training, reducing its motivation to improve. Conversely, when $\alpha$ is too large, the reward curve becomes overly flat in the high-error region, where model's predictions commonly fall in the early stages of training, making it difficult for the model to determine a clear direction for improvement.

## D    PERFORMANCE OF PROMPTING STRATEGY

To validate whether the PTS reasoning pattern can be learned by models via prompting strategies, we design prompting experiments. Specifically, we first formulate the process of perception elaboration and decomposition into a system prompt (presented in the Appendix). Then, we select data from the collected PTS instructions as demonstrations. We use Qwen2.5-VL-7B as the backbone model and

provide it with the system prompt along with varying numbers of demonstrations ($k = 0, 1, 4, 8$) to perform in-context learning. The results are presented in Table 3. In the 0-shot setting, model perform even worse then the direct answer setting, indicating that the highly structured reasoning pattern of PTS can hardly learned by the model through prompting alone. As the number of demonstrations increases, the model's performance improves. However, when $k = 8$, the performance drops again, reflecting that too many demonstrations can lead to confusion due to an overly long context.

# E  ABLATION STUDIES ON SYMBOLIC TOKEN

## E.1  TOKEN DESIGN

We explore various designs for symbolic token styles. Initially, we replace a line segment's symbolic representation with [Dist: xxx]. We also experiment with other abstract tokens, such as #, which is closer to length units, and ? and *, which are unrelated to length. These symbolic tokens are used to replace the original tokens in the SFT data, and we perform the same SFT training. The results are presented in Table 7. The results show that [Dist: xxx] performs relatively poorly compared to the original PTS data. This suggests that perception decomposition alone is insufficient, and that perception elaboration is also necessary for more accurate estimation. Among the other symbolic tokens, # yields the best performance, while * and ? perform worse. We believe this outcome occurs because the shapes of * and ? are more abstract, making it harder for the model to associate them with length or distance.

## E.2  GRANULARITY

We further investigate the impact of token granularity on performance. To do this, we modify the granularity of each = token to represent 0.05, 0.1, 0.2, and 1.0 units, and conduct SFT for each case. The results are shown in Table 8. Notably, when the granularity is too coarse, the model shows relatively better overall estimation accuracy but struggles with high-precision estimation. This highlights the trade-off between granularity and precision in symbolic token representation.

Table 7: Ablation studies on symbolic token designs.

| Symbolic Token | DisTANCE | |
| --- | --- | --- |
| | $RA_{0.1}$ | $RA_{avg}$ |
| = | **16.3** | **45.4** |
| # | 16.7 | 42.9 |
| * | 13.4 | 40.7 |
| ? | 13.0 | 39.6 |
| [DIST:xxx] | 15.7 | 42.0 |

Table 8: Ablation studies on symbolic token granularity.

| Granularity | DisTANCE | |
| --- | --- | --- |
| | $RA_{0.1}$ | $RA_{avg}$ |
| 0.05 | 10.7 | 38.2 |
| 0.1 | **16.3** | 45.4 |
| 0.2 | 12.5 | 44.7 |
| 1.0 | 5.0 | **47.3** |

# F  ABLATION STUDIES ON TRAINING STRATEGY

We analyze the effectiveness of key components in our method. As shown in Table 9, removing any of these components negatively affects the model's performance. Replacing the original exponential reward function with a linear one reduces the model's high-precision accuracy, likely due to its weaker discriminative ability across different relative error levels. Moreover, training without normalizing the ground-truth also results in a performance drop, highlighting the importance of label normalization in stablizing the training process. These results collectively validate the necessity of the designs in PTS. Beyond these ablations, we further experimented with increasing the granularity of symbolic tokens (*e.g.,* using "=" to denote 0.05 units instead of 0.1 units). While this finer division also yields competitive results, performance slightly decreases compared to the 10-division strategy. We hypothesize that 10 equal partitions align more closely with common metric conventions, making them more effective for guiding the model's perception.

Table 9: Ablation studies on key designs in PTS.

| Method | DisTANCE | |
|---|---|---|
| | $RA_{0.1}$ | $RA_{avg}$ |
| Qwen2.5-VL-7B-PTS | **64.7** | **88.3** |
| + symbolic granularity | 62.7 | 83.5 |
| - exponential reward | 60.0 | 83.0 |
| - normalization | 58.0 | 81.5 |

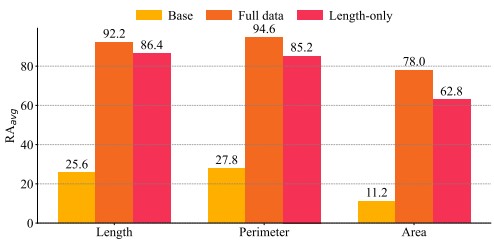

Figure 8: Comparison of the performance of models trained with full/length-only PTS data.

## G DATA EFFICIENCY ANALYSIS

We further investigate the data efficiency of PTS by training models using only length estimation data. As shown in Figure 8, despite being restricted to a single sub-task, the model achieves estimation precision on length, perimeter, and area tasks that is highly comparable to the model trained on the full dataset. This finding suggests that equipping LVLMs with a strong ability in primitive distance estimation enables them to generalize naturally to more complex compositional tasks involving distance aggregation (*e.g.,* perimeter) or multiplication (*e.g.,* area). In particular, length-only training still yields substantial gains over the base model across all sub-tasks, highlighting the central role of length estimation as a foundational skill.

## H COMPUTATIONAL EFFICIENCY ANALYSIS.

We design an efficiency experiment comparing the performance of Qwen2.5-VL-7B, Qwen2.5-VL-7B-CoT, RL-trained Qwen2.5-VL-7B-PTS, and tool-augmented ReVPT-7B (Zhou et al., 2025b) on the DisTANCE dataset. As shown in the table, although the PTS method generates longer sequences, the increase in inference latency is still smaller compared to the tool-based method, highlighting the computational efficiency of our approach.

| Model | Length | Latency |
|---|---|---|
| Qwen2.5-VL-7B | 314.0 | 2.71 |
| Qwen2.5-VL-7B-CoT | 333.1 | 2.85 |
| ReVPT-7B | 360.5 | 6.23 |
| Qwen2.5-VL-7B-PTS | 571.8 | 4.87 |

## I PTS DATA COLLECTION DETAILS

### I.1 INSTRUCTION CASE

Below is an instruction example in the PTS format.

> **PTS Paradigm**
>
> <think>Okay, let me try to figure this out.
>
> <review>
> The problem is to estimate the relative length of the orange triangle's longest side compared to the image's height. To solve it, I need to estimate the lengths of both and divide the length of the orange triangle's longest side by the image's height.
> </review>

<hint>
The task is a precise measurement one. It requires accurately estimating the dimensions within the image. To solve such tasks, I can pick a dimension as the reference and compare other dimensions to it for estimation. To represent length more accurately, I'll use text tokens to visualize the dimensions in the image. The reference line should be 1.0 unit long. I'll represent it as <==========>. The <> are the delimiters and each = is a minimum length unit of 0.1 units long.
</hint>

<reference>
In this case, the dimensions I need to estimate are the orange triangle's longest side and the image's height. Let's use the longest side of the orange triangle as the reference. I define its length as 1.0 unit and represent it as <==========>.
</reference>

<estimation>
Next, I need to estimate the image's height. I visually compare the image's height to the reference line. The image's height seems to be:

First segment <==========> (1.0 unit)
Second segment <=========> (0.9 units)

So the height is 1.0 + 0.9 = 1.9 units.
</estimation>

<calculation>
Now for the math: the length of the triangle's side (1.0) divided by the image height (1.9). 1.0 / 1.9 = 0.526, which rounds neatly to 0.53.
</calculation>

Yeah, that checks out. So the answer should be 0.53.</think>

<answer>0.53</answer>

## I.2 PROMPT FOR SYNTHESIZING PTS INSTRUCTIONS

- For *perimeter* estimation, the PTS chain explicitly computes the length of each polygonal edge before summing them.

- For *area* estimation, the PTS chain first identifies the appropriate geometric formula (*e.g.,* rectangle, triangle, circle) and then estimates the required line segments.

*Prompt for synthesizing PTS instructions*

I would like you to convert a json string into a detailed thought process.

Your thought process should include the following parts:
1. Review the task (enclosed in <review></review>)
2. Define the estimation methods (select reference line and symbolize dimensions as text tokens) (enclosed in <hint></hint>)
3. Select a side as the reference line (enclosed in <reference></reference>)
4. Visually compare other sides to the reference line and estimate their length (enclosed in <estimation></estimation>)
5. Use the necessary formulas to calculate the final result (enclosed in <calculation></calculation>)

> The whole thought process should be formatted as follows:
> <think>[Your thought process]</think>
> <answer>[Final answer]</answer>
>
> Here are some good cases:
> input1:
> output1:
>
> ...
>
> Now, using the following data, please synthesize the thought process. When you mention the information in the json, you MUST pretend that you observe them from an image. Avoid using words like "according to the symbolization result...". No matter how long the line is, you MUST go through every reference segment it contains, instead of using words like "There's a sequence of thirteen <==========> segments".

## J    IMPLEMENTATION DETAILS

### J.1    TRAINING DETAILS OF PTS

We select Qwen2.5-VL-3B/7B as the base model. Both models are first supervised fine-tuned with the 6,000 collected PTS data, then further trained by GRPO on another 6,000 answer-only distance reasoning data. We normalize the ground-truth of the first 3,000 data and leave the other data with random distribution. The $\alpha$ is set to 3 in practice. We utilize the implementation of GRPO in EasyR1 (Zheng et al., 2025), with `rollout_batch_size=512` and `global_batch_size=128`.

### J.2    BASELINES

- SpaceThinker (remyxai, 2025) is trained to enhance spatial reasoning with test-time compute by fine-tuning on synthetic reasoning traces on real-world images with 3D annotations.
- Spatial-R1 (Ouyang, 2025)improves LVLMs' spatial reasoning capabilities by introducing the spatial reasoning dataset and task-specific GRPO fine-tuning.
- DetToolChain (Wu et al., 2024b) introduces a tool-augmented prompting paradigm that equips LVLMs with zero-shot object detection ability. Since the work has not been open-sourced, we reproduced its Ruler Marker strategy by adding scale marks along the edges of the image and take the `gpt-4o-0806` as the base model.
- Visual Sketchpad (Hu et al., 2024) equips LVLMs with a drawing-based reasoning framework, enabling them to sketch lines, boxes, and marks (with optional vision tools) to enhance visual reasoning. We take the `gpt-4o-0806` as the base model and set the `task_type=vision` for evaluation.

## K    QUALITATIVE ANALYSIS

In this section, we compare the generation cases of GPT-4o, Qwen2.5-VL-7B and Qwen2.5-VL-7B-PTS on DisTANCE. The questions and images are presented in Figure 9, 10 and 11.

### K.1    LENGTH ESTIMATION

> **Length estimation case: Qwen2.5-VL-7B-PTS**
>
> <think>Let's tackle this problem step by step.

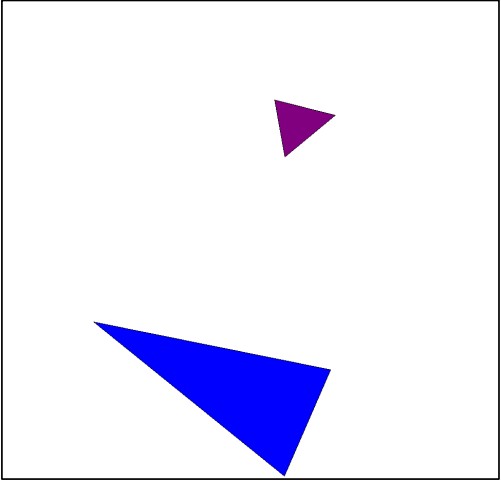

**Question:**
Determine the ratio of the length of the
height of the image to that of the shortest
side of the purple triangle.

**Ground-truth:**
8.21

Figure 9: Length estimation case from DisTANCE.

The objective is to determine the ratio of the height of the image to the shortest side of the purple triangle. To achieve this, I need to ascertain the lengths of these two sides and then compute the ratio by dividing the height of the image by the shortest side of the purple triangle.

This task involves precise measurement, requiring accurate estimation of the dimensions within the image. A strategic approach is to select one dimension as the reference and compare other dimensions against it for estimation. To visually depict the lengths, I'll use text tokens to represent the dimensions in the image. I'll establish the reference line as 1.0 unit long and represent it as <==========>. Here, the <> are delimiters, and each = symbolizes a minimum length unit of 0.1 units.

In this scenario, the two dimensions of interest are the shortest side of the purple triangle and the height of the image. I'll choose the shortest side of the purple triangle as the reference line. Thus, its length is defined as 1.0 unit, represented as <==========>.

Now, I need to estimate the height of the image. By visually comparing the image height to our reference line, it can be broken down into segments as follows:
First segment <==========> (1.0 unit),
Second segment <==========> (1.0 unit),
Third segment <==========> (1.0 unit),
Fourth segment <==========> (1.0 unit),
Fifth segment <==========> (1.0 unit),
Sixth segment <==========> (1.0 unit),
Seventh segment <==========> (1.0 unit),
Eighth segment <==========> (1.0 unit),
And a final segment <====> (0.4 units).
Adding these together, the height of the image is 1.0 + 1.0 + 1.0 + 1.0 + 1.0 + 1.0 + 1.0 + 1.0 + 0.4 = 8.4 units.

For the final calculation, I divide the height of the image (8.4) by the shortest side of the purple triangle (1.0). So, 8.4 × 1.0 = 8.4.
Everything checks out. The answer is 8.4.</think>

<answer>8.4</answer>

---

*Length estimation case: GPT-4o*

To find the ratio of the height of the image to the shortest side of the purple triangle, let's estimate these measurements:

1. Height of the Image: The height of the image is the vertical dimension from the top to the bottom. By visually examining the image, this can be approximated as roughly 800 units based on the pixel dimensions.

2. Shortest Side of the Purple Triangle: By inspecting the purple triangle, it seems that the shortest side is roughly 30 units in length.

Now, calculate the ratio:
$$\text{Ratio} = \frac{\text{Height of the image}}{\text{Shortest side of the purple triangle}} = \frac{800}{30} \approx 26.67$$
Therefore, the ratio is $\boxed{26.67}$.

---

*Length estimation case: Qwen2.5-VL-7B*

To determine the ratio of the height of the image to the shortest side of the purple triangle, we need to estimate the dimensions of the image and the triangle. Let's assume the image is a rectangle and the triangle is a right triangle with the shortest side being the base.

First, let's estimate the dimensions of the image. The image appears to be a rectangle with a height of approximately 10 units and a width of approximately 15 units.

Next, let's estimate the dimensions of the purple triangle. The triangle appears to be a right triangle with the shortest side (the base) being approximately 5 units and the height being approximately 8 units.

The ratio of the height of the image to the shortest side of the purple triangle is:

$$\frac{\text{height of the image}}{\text{shortest side of the triangle}} = \frac{10}{5} = 2$$

Therefore, the ratio is $\boxed{2}$.

## K.2 PERIMETER ESTIMATION

*Perimeter estimation case: Qwen2.5-VL-7B-PTS*

<think>Let's tackle this problem step by step.
The task is to determine the relative length of the perimeter of the image compared to the perimeter of the blue rectangle. Essentially, we need to find out how many times the perimeter of the blue rectangle fits into the perimeter of the image. This will involve estimating the perimeters of both shapes and then dividing the perimeter of the image by that of the blue rectangle.
This task requires precise measurement. It involves accurately estimating the perimeters within the image. To solve such tasks, we can select a dimension as a reference and

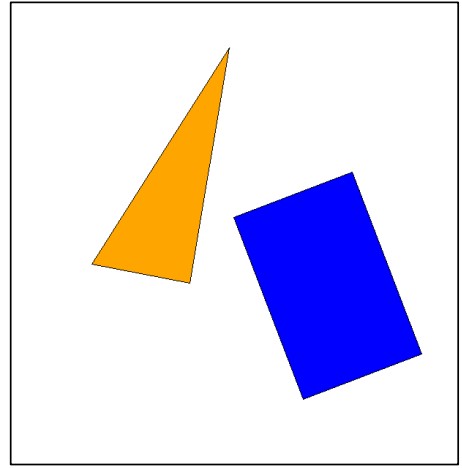

**Question:**
The perimeter of the image is to the
perimeter of the blue rectangle as what is
to 1?

**Ground-truth:**
2.82

Figure 10: Perimeter estimation case from DisTANCE.

compare other dimensions to it for estimation. To represent length accurately, we'll use text
tokens to visualize the dimensions in the image. The reference line should be 1.0 unit long.
We'll represent it as <==========>. The <> are delimiters, and each = is a minimum
length unit of 0.1 units long. For the perimeter of the image, we add up the lengths of all
its sides (width and height multiplied by 2 since it's a rectangle-like shape). For the blue
rectangle, we sum up the lengths of its sides as well.

In this case, let's take the shorter side of the blue rectangle as our reference. I define its
length as 1.0 unit and represent it as <==========>. I'll estimate the lengths of other
sides by comparing them to this reference.

First, I estimate the longer side of the blue rectangle. By visually comparing it to the
reference line, it appears to consist of:

A first segment <==========> (1.0 unit),

And an additional segment <======> (0.6 units).

So, the longer side of the blue rectangle is 1.0 + 0.6 = 1.6 units.

Therefore, the perimeter of the blue rectangle equals 2 * (longer side + shorter side) = 2 *
(1.6 + 1.0) = 5.2 units. I estimate the width of the image. By visually comparing it to the
reference line, it appears to consist of:

A first segment <==========> (1.0 unit),

A second segment <==========> (1.0 unit),

A third segment <==========> (1.0 unit),

And a final segment <=======> (0.7 units).

Adding these together, the width of the image is 1.0 + 1.0 + 1.0 + 0.7 = 3.7 units. I estimate
the height of the image. By visually comparing it to the reference line, it appears to consist
of:

A first segment <==========> (1.0 unit),

A second segment <==========> (1.0 unit),

A third segment <==========> (1.0 unit),

A fourth segment <==========> (1.0 unit).

Adding these together, the height of the image is 1.0 + 1.0 + 1.0 + 1.0 = 4.0 units.
Therefore, the perimeter of the image equals 2 * (width + height) = 2 * (3.7 + 4.0) = 15.4 units.
Finally, the relative length of the perimeter of the image compared to the perimeter of the blue rectangle is 15.4 / 5.2 ≈ 2.96.
That seems correct. So the answer should be 2.96.</think>

<answer>2.96</answer>

---

*Perimeter estimation case: GPT-4o*

To determine the ratio, we need to estimate the perimeters of both the orange triangle and the blue rectangle visible in the image.

1. Estimate the perimeter of the blue rectangle:
- It appears as a standard rectangle. Let's estimate dimensions: assume the height is approximately 1.8 times the width.
- If width $w = x$, then height $h = 1.8x$.
- Perimeter of rectangle $P = 2(w + h) = 2(x + 1.8x) = 3.6x$.

2. Estimate the perimeter of the orange triangle:
- Judging by the image, let's assume side lengths roughly proportionate: $a = x$, $b = 1.5x$, $c = 1.2x$.
- Perimeter $P = a + b + c = x + 1.5x + 1.2x = 3.7x$, compute the ratio of the total perimeter to just the blue rectangle:Ratio $= \frac{\text{Perimeter of the orange triangle}}{\text{Perimeter of the blue rectangle}} = \frac{3.7x}{3.6x} = \frac{3.7}{3.6}$ By calculating the ratio:$\frac{3.7}{3.6} \approx 1.0278$
Hence, the answer is: $\boxed{1.03}$

---

*Perimeter estimation case: Qwen2.5-VL-7B*

To solve this problem, we need to calculate the perimeters of the orange and blue rectangles and then compare them.

First, let's determine the dimensions of the orange rectangle. The orange rectangle is a 3x4 rectangle, so its perimeter is:
$2 \times (3 + 4) = 2 \times 7 = 14$, let's determine the dimensions of the blue rectangle. The blue rectangle is a 5x3 rectangle, so its perimeter is:$2 \times (5 + 3) = 2 \times 8 = 16$, we need to find the ratio of the perimeter of the orange rectangle to the perimeter of the blue rectangle. This ratio is:
$\frac{14}{16} = \frac{7}{8} = 0.875$

Therefore, the final answer is:
$\boxed{0.875}$

## K.3 AREA ESTIMATION

*Area estimation case: Qwen2.5-VL-7B-PTS*

<think>Alright, let's start working this out.
The task is to calculate how large the image is relative to the purple circle in terms of area. To achieve this, I need to estimate the areas of both the image and the purple circle, and then divide the area of the image by the area of the purple circle.
This task demands precise measurement, requiring accurate estimation of areas within the

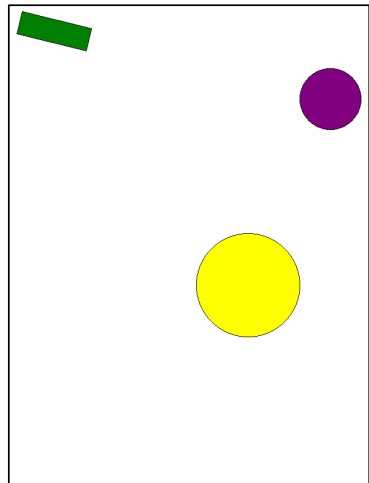

**Question:**
I need you to calculate how large the image
is relative to the purple circle in terms of
area.

**Ground-truth:**
58.64

Figure 11: Area estimation case from DisTANCE.

image. I can select a dimension as a reference and compare other dimensions to it for estimation purposes. To represent lengths more precisely, I'll use text tokens to visualize the dimensions in the image. The reference line should be set as 1.0 unit long and represented as <==========>. The <> are delimiters, and each = represents a minimum length unit of 0.1 units. For the area of the image, since it has a rectangle-like shape, we calculate it by multiplying its width and height. For the circle, we use the formula for the area of a circle, which is A = ŏ3c0rŏ0b2, where r is the radius. So, I need to estimate the radius of the purple circle, the width, and the height of the image.
In this case, let's take the radius of the purple circle as our reference. I'll define its length as 1.0 unit and represent it as <==========>. I will estimate the lengths of other sides by making visual comparisons to this reference line.
First, I estimate the width of the image. When visually comparing the image width to our reference line, I can break it down as follows:
There's the first <==========> segment (1.0 unit),
A second <==========> segment (1.0 unit),
A third <==========> segment (1.0 unit),
A fourth <==========> segment (1.0 unit),
A fifth <==========> segment (1.0 unit),
A sixth <==========> segment (1.0 unit),
A seventh <==========> segment (1.0 unit),
An eighth <==========> segment (1.0 unit),
A ninth <==========> segment (1.0 unit),
A tenth <==========> segment (1.0 unit),
An eleventh <==========> segment (1.0 unit),
A twelfth <==========> segment (1.0 unit),
And finally, a <=====> segment, which is 0.5 units long.
The total width of the image sums up to 1.0 * 12 + 0.5 = 12.5 units.
, I estimate the height of the image. By visually comparing the image height to our reference line, I can break it down like this:

There's the first <===========> segment (1.0 unit),
A second <==========> segment (1.0 unit),
A third <==========> segment (1.0 unit),
A fourth <==========> segment (1.0 unit),
A fifth <==========> segment (1.0 unit),
A sixth <==========> segment (1.0 unit),...(Omitted due to space limitations.)
A thirteenth <==========> segment (1.0 unit),
A fourteenth <==========> segment (1.0 unit),
A fifteenth <==========> segment (1.0 unit),
And finally, a <==> segment, which is 0.2 units long.
The total height of the image sums up to 1.0 * 15 + 0.2 = 15.2 units.
Consequently, the area of the purple circle is calculated using the formula $A = \pi r^2$, with r
= 1.0 and $\pi$ approximately 3.14. So, the area is $3.14 * 1.0^2 = 3.14$ square units.
For the image, since we know the width is 12.5 units and the height is 15.2 units, using the
formula for the area of a rectangle A = width * height, the area of the image is 12.5 * 15.2
= 190.0 square units.
Finally, the proportion of the image's area relative to the purple circle's area is 190.0 / 3.14
$\approx$ 60.7.
Yes, that seems right. So the answer should be 60.7.</think>
<answer>60.7</answer>

---

*Area estimation case: GPT-4o*

To calculate the area of the image relative to the purple circle, we need to estimate the
radius of the purple circle and the dimensions of the image first.

Let's start by estimating the radius of the purple circle. Based on the image:
- The purple circle appears small in comparison to the image dimensions.

Let's estimate the purple circle's radius as follows:
- Assume the radius of the purple circle ($r$) consists of roughly 1/10th of the width of the
image.
- The image width appears to be in a ratio of approximately 4:3 (width:height).
Now, let's calculate the area of the image and the area of the circle:
Assume the image width is 800 pixels and height is 600 pixels (which are typical
dimensions for the aspect observed).
- Image area = $width \times height = 800 \times 600 = 480,000$ pixels$^2$.

Assuming the estimated radius of the purple circle:
- $r = \frac{800}{10} = 80$ pixels.

Circle area = $\pi r^2 = \pi \times 80^2 = 6,400\pi$ pixels$^2$.
Now calculate the relative area:
- Relative area = $\frac{Image Area}{Circle Area} = \frac{480,000}{6,400\pi} \approx \frac{480,000}{20,096} \approx 23.89$.

The relative area of the image compared to the purple circle is approximately: 23.89

---

*Area estimation case: Qwen2.5-VL*

To estimate the size of the image relative to the purple circle in terms of area, we need to
compare the areas of the two shapes. However, since the image is not provided, I will use a
hypothetical scenario where the purple circle is a standard circle with a radius of 1 unit for
simplicity.
1. Area of the Purple Circle:
The area $A$ of a circle is given by the formula:

$$A = \pi r^2$$

For a circle with radius $r = 1$:

$$A = \pi \times 1^2 = \pi \approx 3.14$$

2. Estimating the Area of the Image:
The image appears to be a rectangle with a width and height that can be estimated. Let's assume the width and height of the rectangle are both approximately 2 units each (this is a rough estimate based on the visual proportions).
The area $A$ of a rectangle is given by:

$$A = \text{width} \times \text{height}$$

For a rectangle with width and height both 2 units:

$$A = 2 \times 2 = 4$$

3. Comparing the Areas:
The area of the image (rectangle) is approximately 4 square units, and the area of the purple circle is approximately 3.14 square units.
4. Ratio of the Areas:
The ratio of the area of the image to the area of the purple circle is:

$$\text{Ratio} = \frac{\text{Area of the Image}}{\text{Area of the Purple Circle}} = \frac{4}{3.14} \approx 1.27$$

Therefore, the image is approximately 1.27 times the area of the purple circle. $\boxed{1.27}$

## L  QUALITATIVE ANALYSIS ON GENERAL TASKS

### L.1  CHART UNDERSTANDING

---

*Generation case of PTS*

The graph shows the value of bread, biscuits, wafers, cakes, and pastries exported from the United Kingdom (UK) from 2001 to 2019. To determine if 2010 was the year with the most value, we need to compare the values for each of the given years.

The task is to visually assess the data and identify the year with the highest value. We can achieve this by selecting a reference line and comparing other values to it. I'll use text tokens to represent the lengths visually. The reference line will be 1.0 unit long, depicted as <==========>. The <> are delimiters, and each = represents a minimum length unit of 0.1 units.

For this task, the relevant dimensions are the years and the corresponding values. I'll select the value for 2010 as the reference line, defining it as 1.0 unit long, represented as <==========>.

Now, let's estimate the values for other years by comparing them to the reference line:

- 2005: The value appears to be <========>, which is 0.8 unit.
- 2015: The value is <==========><===>, which is 1.3 units.
- 2019: The value is <==========><=====>, which is 1.5 units.

---

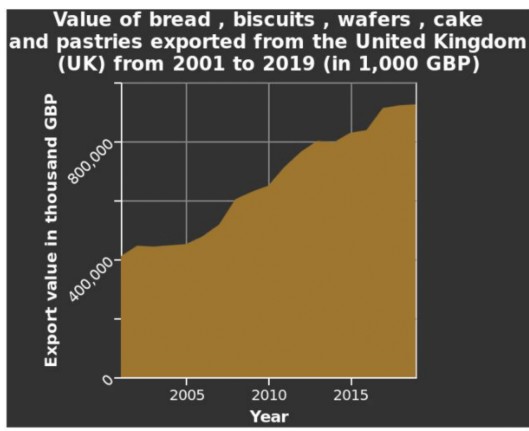

**Question:**
From 2001 to 2019, Is 2010 exported the most value of
bread, biscuits, wafers, cakes and pastries from the UK?

**Ground-truth:**
No

Figure 12: Case of chart understanding.

The values for 2015 and 2019 are higher than the reference line, indicating they are the most valuable. However, the question asks for the value in 2010, which is already defined as 1.0 unit, and visually, it matches the reference line.

So, the final answer is that 2010 is one of the years with the most value, but it is not the highest. The actual answer should be that 2019 has the most value.

## L.2 FINE-GRAINED PERCEPTION

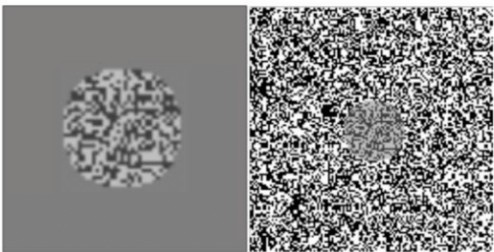

**Question:**
The image shows the famous Chubb illusion. Are the
center areas of two rectangular fields identical?

**Ground-truth:**
No

Figure 13: Case of fine-grained perception.

---

*Generation case of PTS*

The task is to determine if the center areas of two rectangular fields are identical. To solve this, I need to compare the two areas visually and assess their equivalence. The approach involves selecting a reference dimension and comparing other dimensions to it for estimation purposes. I'll use text tokens to represent the lengths, with a reference line set to 1.0 unit long, depicted as <==========>. The delimiters <> and each = represent a minimum length unit of 0.1 units long. For the areas, since they are rectangular, we calculate the area by multiplying their lengths. Therefore, I need to estimate the lengths of the sides of both rectangles.

In this scenario, let's take the shorter side of the first rectangle as our reference. I define its length as 1.0 unit and represent it as <==========>. I will estimate the lengths of other sides by comparing them to this reference.

First, I estimate the longer side of the first rectangle. By visually comparing it to the reference line, it appears to consist of:
One segment <==========> (1.0 unit),
And an additional segment <======> (0.6 units).
So, the total length of the longer side of the first rectangle is 1.0 + 0.6 = 1.6 units.
Now, I estimate the longer side of the second rectangle. By visually comparing it to the reference line, it seems to be composed of:
One segment <==========> (1.0 unit),
And another segment <========> (0.8 units).
Adding these together, the total length of the longer side of the second rectangle is 1.0 + 0.8 = 1.8 units.
Next, I estimate the shorter side of the second rectangle. By visually comparing it to the reference line, it appears to consist of:
One segment <==========> (1.0 unit),
And an extra segment <==> (0.2 units).
So, the total length of the shorter side of the second rectangle is 1.0 + 0.2 = 1.2 units.
For the first rectangle, using the formula for the area of a rectangle A = length * width, the area is 1.6 * 1.0 = 1.6 square units.
For the second rectangle, using the same formula, the area is 1.8 * 1.2 = 2.16 square units.
Since the calculated areas are different (1.6 and 2.16), the center areas of the two rectangular fields are not identical.
The answer is that the center areas of the two rectangular fields are not identical.

---

### L.3 TOP-DOWN VIEW UNDERSTANDING

---

*Generation case of PTS*

Let's tackle this step by step.
The objective is to determine which of the two, the master bedroom or the patio, is larger in terms of area. To do this, I need to estimate the areas of both the master bedroom and the patio and then compare them. This task requires precise measurement. It involves accurately estimating the areas within the image. To solve such tasks, I can select a dimension as a reference and compare other dimensions to it for estimation purposes. To represent lengths more precisely, I'll use text tokens to visualize the dimensions in the image. The reference line should be set as 1.0 unit long, represented as <==========>. The <> are delimiters, and each = represents a minimum length unit of 0.1 units. For the area of the rectangle-like shapes, we calculate it by multiplying the length and width. So, I need to estimate the lengths of the dimensions of both the master bedroom and the patio.
In this case, let's take the shorter side of the master bedroom as our reference. I'll define its length as 1.0 unit and represent it as <==========>. I will estimate the lengths of other sides by comparing them to this reference.
First, I estimate the longer side of the master bedroom. By visually comparing it to the reference line, it seems to consist of:

---

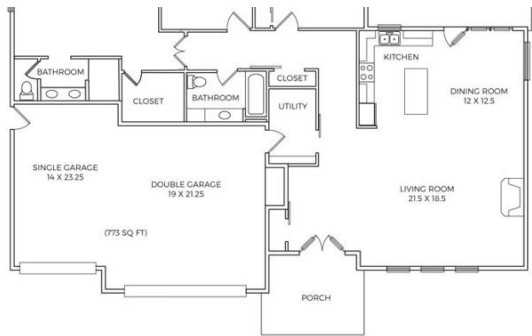

**Question:**
Which room is bigger, the master bedroom or the patio?

**Ground-truth:**
The master bedroom

Figure 14: Case of top-down view understanding.

> The first segment <==========> (1.0 unit),
> And an additional segment <=========> (0.9 units).
> So, the total length of the longer side of the master bedroom is 1.0 + 0.9 = 1.9 units.
> Now, I estimate the longer side of the patio. By visually comparing it to the reference line, it appears to consist of:
> The first segment <==========> (1.0 unit),
> A second segment <========> (0.8 unit),
> Adding these together, the total length of the longer side of the patio is 1.0 + 0.8 = 1.8 units.
> Next, I estimate the shorter side of the patio. By visually comparing it to our reference line, it seems to be composed of:
> One segment <=========> (0.9 unit),
> So, the total length of the shorter side of the patio is 0.9 units.
> Using the formula for the area of a rectangle A = length * width, the area of the master bedroom is 1.9 * 1.0 = 1.9 square units.
> Similarly, using the same formula, the area of the patio is 1.8 * 0.9 = 1.62 square units.
> Finally, to determine which is larger, we compare the areas. The area of the master bedroom (1.9) is greater than the area of the patio (1.62). Therefore, the master bedroom is bigger.
> That checks out correctly. So, the answer is the master bedroom.

## M  RELATED WORK

**Multimodal Reasoning Models.**   Inspired by the success of inference-time scaling in large language models (Tang et al., 2025b; Wang et al., 2025b), recent research has delved into enhancing the reasoning capabilities of Large Vision-Language Models (LVLMs) (Bai et al., 2025; Zhu et al., 2025; Wu et al., 2025; Zhan et al., 2025b). Early methods (Xu et al., 2024; Du et al., 2025; Yao et al., 2024) focus on collecting multimodal Chain-of-Thought data to conduct supervised fine-tuning on LVLMs. More recently, spurred by the achievements of reinforcement learning algorithms such as GRPO (Shao et al., 2024), some studies have explored the application of reinforcement learning to improve the long-sequence reasoning ability of LVLMs. Some work  (Chen et al., 2025b; Liu et al., 2025c; Zhan et al., 2025a; Zhou et al., 2025a) investigate the design of different verifiable reward functions for various visual tasks. MM-Eureka (Meng et al., 2025) attempts to directly employ rule-based reinforcement learning to stimulate the model's reasoning and reflective abilities without performing SFT. VLAA-Thinker (Chen et al., 2025a) explores the influence of cold start to the reasoning capabilities of LVLMs. Our work focuses on adopting RL to enable LVLMs to solve precise measurement problems by including their perception process into text-formed long

reasoning chains. More recently, some work (Wang et al., 2025a; Zhou et al., 2025b; Yang et al., 2025a) introduce rethinking reasoning chains and agentic tool calling method to further enhance LVLMs performance on complex multimodal tasks.

**Visual Perception in LVLMs.** As LVLMs achieve impressive progress in multimodal reasoning tasks, their visual perception accuracy remains a key bottleneck to their real-world applications. Some work studies the hallucination issue by evaluating whether LVLMs accurately describe image contents like object class (Li et al., 2023) and relationship (Wu et al., 2024a). HallusionBench (Guan et al., 2024) further challenges LVLMs with detailed visual illusions, while MMVP (Tong et al., 2024b) evaluates LVLMs on image pairs that share similar visual representations but significant semantic information. Recently, spatial intelligence (Chen et al., 2024) has attracted increasing attention. Existing works (Yang et al., 2024; Cheng et al., 2024) have proposed benchmarks to evaluate the spatial reasoning abilities of LVLMs, including understanding relative spatial relationships and estimating absolute distances. Unlike real-world estimation benchmarks, DisTANCE consists of 2D synthetic geometric shapes, eliminating the need for commonsense knowledge and concentrating solely on the model's ability to perceive basic distance information.

