# OpenReview forum: "Unleashing Perception-Time Scaling to Multimodal Reasoning Models"
_ICLR.cc/2026/Conference — ICLR 2026 Poster_

### Official Review · Reviewer_EEJH · 2025-10-27

**Soundness:** 3
**Presentation:** 3
**Contribution:** 3
**Rating:** 8
**Confidence:** 3

**Summary:**

The paper introduces a new method for inference-time scaling for perception-heavy reasoning tasks such as visual estimation tasks (i.e. tasks that require estimating the size of geometric elements). The paper introduces a new way for inference-time scaling called perception-time-scaling (PTS). The main idea is that models are encouraged to decompose tasks by using some tokens for "unit estimation" (i.e.  <==========> corresponds to 1 unit). To encourage models to use this method, they introduce a task Distance reasoning Task with Analytical Numeric and Comparative Estimation (DisTANCE), which is automatically generated using programmatically generated images which includes a known ground truth, and PTS (and CoT for ablations) reasoning traces using GPT4o. Combining SFT and RL they train models and show that PTS leads to better performance than CoT. They also show that finetuning on this data, improves performance for other perception heavy tasks such as Geoperception and Lego-Puzzles.

**Strengths:**

- The method is innovative and addresses a well-recognized challenge in VLLMs.
- The ablation study comparing CoT and PTS (Tables 3 and 4) is convincing: SFT+RL with PTS yields notable gains over SFT+RL with CoT, with improvements exceeding 10 accuracy points on DisTANCE, though the benefits on out-of-distribution tasks are more modest.
- The appendix provides a comprehensive set of ablations that further motivate and support the proposed approach.

**Weaknesses:**

While the method appears effective for perceptual measurement tasks, its usefulness in out-of-domain settings is less convincing. Beyond measurement tasks, the gains seem limited at best. For example, in Table 5 the improvements are marginal, and the absence of multiple runs or error bounds further reduces confidence in their significance.

**Questions:**

- In Table 4, why does the method yield larger improvements on GeoLHC for Qwen2.5VL-3B, but greater gains on LegoHeight for the 7B model? Reporting results over multiple runs with deviations would help clarify this.
- In Figure 4b, which tasks are used to evaluate image-focused attention? Is this analysis specific to DisTANCE, or do the observed patterns generalize to out-of-domain tasks as well?

---

> ### Author Response · Authors · 2025-11-23
> **Response to Reviewer EEJH**
>
> Thank you for your positive feedback and insightful comments! We truly appreciate the time you took to review our paper. Below, we address each of your points in detail. We hope these responses help clarify our paper and address your concerns. Additionally, we have made corresponding changes in the PDF, with relevant sections highlighted in *light red*.
>
> > Q1: Why does the method yield larger improvements on GeoLHC for Qwen2.5VL-3B, but greater gains on LegoHeight for the 7B model? Reporting results over multiple runs with deviations would help clarify this.
>
> Thanks for your suggestion! We run each checkpoint five times and includ the standard deviations, but the results still follow the same pattern:
>
> | Model                | Geo_LHC      | LEGO_Height  |
> | -------------------- | ------------ | ------------ |
> | Qwen2.5-VL-3B-PTS-RL | 74.65 ± 1.58 | 41.84 ± 2.06 |
> | Qwen2.5-VL-7B-PTS-RL | 79.79 ± 1.84 | 34.21 ± 1.46 |
>
> We believe this phenomenon arises because the performance of LVLMs on perception tasks is not strongly correlated with model size. Previous studies [1,2] suggest that the language prior in LVLMs can interfere with the accurate perception of visual content, and this interference becomes more pronounced as the model size increases. Since both datasets we evaluated are purely vision-dependent,  it is not surprising that the smaller 3B model outperforms the larger 7B model on one of them.
>
> > Q2: which tasks are used to evaluate image-focused attention? Is this analysis specific to DisTANCE, or do the observed patterns generalize to out-of-domain tasks as well?
>
> The attention analysis is based on the model's generated results from both DisTANCE and the out-of-domain datasets in Table 4. Therefore, we believe this pattern is generalizable.
>
> **References**
>
> [1] Evaluating Object Hallucination in Large Vision-Language Models
>
> [2] Mitigating Object Hallucinations in Large Vision-Language Models through Visual Contrastive Decoding

---

> > ### Comment · Reviewer_EEJH · 2025-11-27
> >
> > Thank you for your response. I maintain my score.

---

> > > ### Author Response · Authors · 2025-11-27
> > >
> > > Thank you for your positive evaluation and for acknowledging the value of our work. We truly appreciate your feedback!

---

### Official Review · Reviewer_eNuB · 2025-10-27

**Soundness:** 2
**Presentation:** 2
**Contribution:** 2
**Rating:** 4
**Confidence:** 4

**Summary:**

This paper introduces DisTANCE, a new perception-centric benchmark for visual estimation tasks. The authors show that most VLMs struggle to accurately assess visual perception capabilities. To address this limitation, they propose the Perception-Time Scaling (PTS) training paradigm, which consists of two stages: (1) a cold-start SFT phase using PTS-annotated data, and (2) a GRPO optimization phase. This approach achieves significant performance gains, improving accuracy by 8.0% to 64.7% over baseline models.

**Strengths:**

- This paper introduces a new perception-centric benchmark and demonstrates that even state-of-the-art proprietary VLMs exhibit limited performance, highlighting the importance of enhancing not only reasoning but also perception capabilities.

- The paper also presents the Perception-Time Scaling (PTS) paradigm, which enables small VLMs to learn to perceive images in a more human-like manner.

**Weaknesses:**

- One of the main contributions is the introduction of a new perception-centric benchmark, particularly focusing on visual estimation, which differs from prior perception tasks. To further enhance the value of this benchmark, I recommend evaluating proprietary reasoning models (e.g., Gemini-2.5-Pro). Since the benchmark contains only 300 samples, this additional experiment seems feasible. When I tested one of the examples from the paper using GPT-5, the model invoked tool-calling behavior and solved the task successfully. However, please note that this comment does not affect my rating of the paper.

- The term Inference-Time Scaling usually refers to the phenomenon where allocating more inference-time compute leads to improved performance. However, the proposed Perception-Time Scaling seems somewhat different in nature. From my understanding, it adds perception-related steps into the reasoning process during training—so it appears closer to “perception-enhanced reasoning.” Moreover, the experiments do not show how performance changes when scaling up perception-related compute during inference. As a result, the naming feels slightly forced, as if it were attached to align with the popular term “inference-time scaling.”

- In the perception elaboration and perception decomposition stages, symbolic tokens (e.g., <==========>) are used. I am curious how performance changes if these tokens are simply provided as prompt hints during inference. This could clarify whether the model truly lacked visual estimation capability and whether reinforcement learning genuinely boosted this ability.

- If the benchmark was generated programmatically in Python, scaling up the dataset for training should be relatively straightforward. It would be interesting to see whether simply training on a larger dataset of similar style improves performance. Such a setting could help demonstrate that data scaling alone is insufficient for the proposed visual estimation task, reinforcing the task’s importance. That said, I understand that conducting this experiment within the rebuttal period may be unrealistic—so even a qualitative discussion or expected outcome would be helpful.

- Intuitively, I expected tool-augmented RL-trained VLMs to perform better. However, I suspect that your proposed method offers significant efficiency advantages. Could you provide details on the actual inference speed or runtime efficiency?

- Could you clarify the decoding settings (e.g., top-p, temperature) used during inference?

- GRPO performance can be highly sensitive to reward design. I am curious why the reward was defined in a continuous form. For instance, what would happen if you used a binarized reward—assigning a reward of 1 when the estimated value difference is below 0.1, and 0 otherwise? A detailed ablation analysis on this design choice would be insightful, especially since recent work [1, Fig. 6] reports that binary rewards can yield better results.

- While I recognize the substantial experimental effort behind this work, the methodological novelty appears limited. The proposed approach mainly adapts GRPO to the visual estimation task rather than introducing new RL algorithms. This is not a major issue, but it does make the paper feel more like an application of existing methods rather than a conceptual advance. I would have appreciated more insightful discussions, such as an analysis of the trade-off between perception and reasoning. If possible, could you elaborate on potential directions or reflections in this regard? For example, which part should we consider when we empower VLM's perception capability? should we strengthen VLM's reasoning part for better perception?


Additional Minor Suggestions (Not Affecting the Score)

- The term “Fast Perception” feels somewhat ambiguous. Would “Fast Thinking” be a more appropriate description?

- In line 187, the result shown in Figure 2 seems more like a fine-grained analysis than a decomposition of a complex perception task—so the phrase “decompose complex perception” was a bit confusing.

- In the Appendix section “Prompt for synthesizing PTS instructions”, a newline character (\n) seems to be missing between items 4 and 5.

---

Reference

[1] Unified Reinforcement and Imitation Learning for Vision-Language Models.

**Questions:**

Detailed questions in the weaknesses part.

---

> ### Author Response · Authors · 2025-11-23
> **Response to Reviewer eNuB [1/4]**
>
> Thank you for your thoughtful feedback and insightful comments. We truly appreciate the time you took to review our paper. Below, we address each of your points in detail. We hope these responses help clarify our paper and address your concerns. Additionally, we have made corresponding changes in the PDF, with relevant sections highlighted in *light green*.
>
> > Q1: Can author evaluate advanced proprietary reasoning models like GPT-5 and Gemini-2.5-Pro?
>
> Sure! The performance of these powerful proprietary models will more comprehensively reflect the difficulty of the proposed DisTANCE. We evaluate GPT-4.1, GPT-5, and Gemini-2.5-Pro and present the evaluation results in the table below and in Table 1 of the PDF.
>
> | Model          | Length | Perimeter | Area | Average |
> | -------------- | ------ | --------- | ---- | ------- |
> | GPT-4.1        | 63.2   | 55.4      | 33.2 | 50.6    |
> | GPT-5          | 72.6   | 72.8      | 46.8 | 64.1    |
> | Gemini-2.5-Pro | 67.8   | 69.2      | 50.4 | 62.5    |
>
> The results show that these more powerful models indeed achieved better performance.
>
> > Q2: Is the term 'Perception-Time Scaling' appropriate, given that it seems more akin to 'perception-enhanced reasoning' during training, rather than scaling perception-related compute during inference as implied by the term?
>
> It is indeed true that our approach does not directly scaling perception-related computation (e.g., in visual encoders). Instead, we use symbolic tokens during inference to guide the model's perception. These tokens represent perceptual information about distances, which can be viewed as the model "perceiving" the image step-by-step. From this perspective, the generation of symbolic tokens during inference introduces additional perception-related steps. Thus, although our approach does not scale compute in the traditional sense, it effectively scales the perception-related tokens in the whole reasoning process. This is why we named the method "Perception-Time Scaling". It emphasizes the extension of perception throughout the entire process, not just during the initial input processing. We hope this explanation clarifies the rationale behind the term.
>
> > Q3: How performance changes if the pattern of symbolic tokens are provided as prompts to the model?
>
> This is a great point. To address it, we designed prompting experiments for validation. Specifically, we first formulate the process of perception elaboration and decomposition into a system prompt. Then, we select data from the collected PTS instructions as demonstrations. We use Qwen2.5-VL-7B as the backbone model and provide it with the system prompt along with varying numbers of demonstrations (k = 0, 1, 4, 8) to perform in-context learning. The results are presented in the table below, as well as in Table 3 of the PDF.
>
> | Model  | Length | Perimeter | Area | Average |
> | ------ | ------ | --------- | ---- | ------- |
> | 0-shot | 12     | 19.8      | 13.4 | 15.1    |
> | 1-shot | 17.8   | 23.2      | 16.2 | 19.1    |
> | 4-shot | 25.8   | 26.0      | 13.6 | 21.8    |
> | 8-shot | 16.6   | 15.8      | 15.6 | 16.0    |
>
> In the 0-shot setting, model perform even worse then the direct answer setting, indicating that the highly structured reasoning pattern of PTS can hardly learned by the model through prompting alone. As the number of demonstrations increases, the model's performance improves. However, when $k = 8$, the performance drops again, reflecting that too many demonstrations can lead to confusion due to an overly long context.
>
> This phenomenon aligns with our earlier observations. Without training, simply providing prompts makes it difficult for the model to learn the structured reasoning process of PTS. This is why we use the two-stage training strategy, where we first allow the model to learn the reasoning pattern through SFT and then further optimize the estimation accuracy through RL.

---

> ### Author Response · Authors · 2025-11-23
> **Response to Reviewer eNuB [2/4]**
>
> > Q4: Would training on a larger dataset of similar style improve performance?
>
> Thank you for your suggestion! We agree that training on datasets of different sizes could help highlight the limitations of data scaling alone for the visual estimation task. While the images are generated using Python code, annotations for CoT and PTS are created with GPT-4o, which is challenging to scale during the rebuttal phase. Therefore, we show the data scaling trend on CoT and PTS using existing data by setting the training data to 2k, 4k, and 6k. Additionally, we scale an extra 6k Direct data. We use these data for the SFT training and summarize the results in the table below and in the PDF.
>
> | Model      | Length | Perimeter | Area | Average |
> | ---------- | ------ | --------- | ---- | ------- |
> | Direct-2k  | 36.8   | 42.6      | 25.8 | 35.1    |
> | Direct-4k  | 41.0   | 56.2      | 30.8 | 42.7    |
> | Direct-6k  | 49.4   | 63.0      | 33.4 | 48.6    |
> | Direct-12k | 55.2   | 68.3      | 36.3 | 53.3    |
> | CoT-2k     | 33.6   | 36.6      | 25.6 | 31.9    |
> | CoT-4k     | 39.2   | 41.6      | 31.8 | 37.5    |
> | CoT-6k     | 40.8   | 52.2      | 29.2 | 40.7    |
> | PTS-2k     | 28.4   | 37.2      | 20.2 | 28.6    |
> | PTS-4k     | 36.6   | 52.4      | 25.8 | 38.3    |
> | PTS-6k     | 44.2   | 62.0      | 30.2 | 45.4    |
>
> Initially, scaling the data results in noticeable improvements, but as the dataset size increases, the performance gains gradually plateau. This indicates that simply scaling the data is insufficient for visual estimation tasks.
>
> > Q5: Could author provide details on the actual inference speed or runtime efficiency?
>
> This is also a good question. I agree that, compared to tool-based methods, our approach offers an advantage by extending the model's reasoning chain to achieve finer-grained perception. Intuitively, the time overhead introduced by the increased sequence length in our method is smaller than the overhead associated with tool invocation. To validate this, we design an efficiency experiment comparing the performance of Qwen2.5-VL-7B, Qwen2.5-VL-7B-CoT, RL-trained Qwen2.5-VL-7B-PTS, and tool-augmented ReVPT-7B (post-trained from Qwen2.5-VL-7B) on the DisTANCE dataset. We focused on two metrics: average sequence length and inference latency. The results are shown in the table below and in the PDF.
>
> | Model             | Sequence Length | Inference Latency |
> | ----------------- | --------------- | ----------------- |
> | Qwen2.5-VL-7B     | 314.0           | 2.71              |
> | Qwen2.5-VL-7B-CoT | 333.1           | 2.85              |
> | ReVPT-7B          | 360.5           | 6.23              |
> | Qwen2.5-VL-7B-PTS | 571.8           | 4.87              |
>
> As shown in the table, although the PTS method generates longer sequences, the increase in inference latency is still smaller compared to the tool-based method, highlighting the computational efficiency of our approach.
>
> > Q6: Could you clarify the decoding settings (e.g., top-p, temperature) used during inference?
>
> Sure, we set the temperature = 0.5 and top_p = 1.0 for all models.

---

> ### Author Response · Authors · 2025-11-23
> **Response to Reviewer eNuB [3/4]**
>
> > Q7: Why not use a binary reward?
>
> This is a good question. We use a continuous reward due to the nature of visual estimation tasks. Unlike typical VQA tasks, which have deterministic answers, visual estimation is more akin to a regression task where the solution space is a continuous range of positive real numbers. Therefore, we need a continuous metric (such as relative error) to assess the quality of a response. It is natural to use a continuous function to provide the reward signal to the model.
>
> Regarding the binary reward based on relative error, it has a key issue: sparse rewards. If we set a high threshold, like 0.1, only very accurate predictions will receive a positive reward. However, in the early stages of training, most predictions are inaccurate, leading to mostly zero rewards. In the context of GRPO, this corresponds to negative advantages, which significantly diminishes the usefulness of the training data, as most of the batch contains negative samples, with few offering positive learning signals. This issue has been discussed in reinforcement learning analysis papers (e.g., DAPO [1]). On the other hand, setting a lower threshold, like 0.5, reduces reward differentiation. The model would then receive the same reward for any prediction within a reasonable range, which confuses the model and makes it more difficult to optimize effectively. For these reasons, we chose a continuous reward. To encourage the model to make more accurate estimates, we used an exponential curve to provide higher gradients in the high-accuracy range.
>
> Regarding the point raised in [2], we believe there is a fundamental difference in our scenario. The reward in [2] consists of two components: the similarity reward, which measures the model's generation style, and the answer reward, which evaluates the correctness of the answer. The similarity reward uses a binary format to avoid ambiguity in comparing styles. In contrast, our reward focuses solely on the correctness of the answer, not the style. We acknowledge that it may be difficult to accurately represent the model's generation style with a continuous score, and in such cases, a binary reward might be a better choice. However, for continuous answers, a smaller relative error directly indicates a higher-quality response. A binary reward would assign the same reward to answers of varying quality, which introduces ambiguity. Therefore, we believe a continuous reward is better suited for our task.
>
> To further validate our point, we used an unfinetuned Qwen2.5-VL-7B as the base model and conducted RL training with binary_reward_0.1, binary_reward_0.5, and our exponential reward. We compared their relative error curves on the validation set (presented in the PDF). The trend aligns with our previous analysis. The proposed continuous exponential reward converges faster and achieves the lowest final relative error.

---

> ### Author Response · Authors · 2025-11-23
> **Response to Reviewer eNuB [4/4]**
>
> > Q8: More discussion about potential directions on the relation of perception and reasoning.
>
> I am glad to share my thoughts on this topic. In our current work, we made a key observation: the perception process can be integrated into reasoning, specifically by allowing the model to represent distance information through symbolic tokens. This approach, combined with reinforcement learning training, significantly enhances the model's perceptual accuracy. Building on this result, we have some thoughts on the future of perception in LVLMs. Specifically, we believe the paradigm of perception needs to shift. Currently, perception is an understanding process , where the model passively describes content from the visual input. We argue that perception should evolve into a generative process, where the model can actively construct representations of what it perceives.
>
> To draw an analogy with human perception: humans not only interpret visual information but also imagine it. This imaginative process is a form of generative perception. We believe that future LVLMs must possess this ability as well. For example, when faced with a complex task that requires transforming an image (e.g., navigating a maze or solving a Sudoku puzzle), generating intermediate visual results is crucial for solving the problem. Additionally, tasks involving spatial transformation require the model not only to understand the original image but also to imagine variations of it, such as changes in perspective. These scenarios demand that the model does more than just understand; it must also generate what it perceives.
>
> Our work can be seen as an initial step toward this generative paradigm through the PTS framework. PTS enables the model to generate symbolic tokens to represent its perception of distance. Although this exploration is still in its early stages, focusing primarily on abstract symbolic tokens and distance-related information, we envision a future where models can generate more semantically rich perceptual content. The content generated by the model would no longer be limited to abstract text tokens, and the tasks it can handle would extend beyond estimation tasks. With the advancement of generative foundation models, we may enable models to call external generative tools to produce intermediate perceptual results. Alternatively, latent reasoning methods could allow models to autonomously generate latent visual tokens [3]. Another key advantage of generative perception is that it further unifies perception and reasoning, enabling them to be jointly optimized during the training process..
>
> In summary, we believe that the future of perception will follow a generative paradigm, closely integrated with reasoning.
>
> > Q9: Fast perception or fast thinking?
>
> We choose the term "Fast Perception" because we observed that the model's generated content can be divided into reasoning-related and perception-related parts. By using "slow-reasoning" and "fast-perception", we want to highlight the different emphasis the model places on reasoning and perception.
>
> > Q10: Clarification on analysis of Figure 2 and "decompose complex perception"
>
> Let us clarify further. Figure 2 shows the model's performance, where we observe that as task difficulty increases, the model's performance decreases. Notably, in visual estimation tasks, more difficult problems are essentially the sum of several simpler ones (e.g., estimating a long segement is equivalent to estimating multiple shorter segments and summing them). Based on this observation, we find that if the models can learn to decompose tasks, its estimation accuracy will improve compared to direct estimation. This is why we emphasize that the current models fail to "decompose complex perception", and introduce the perception decomposition strategy later.
>
> > Q11: Missing '\n' in the case
>
> Thank you for your detailed proof reading! We have fixed it in the PDF.
>
> **References**
>
> [1] DAPO: An Open-Source LLM Reinforcement Learning System at Scale
>
> [2] Unified Reinforcement and Imitation Learning for Vision-Language Models
>
> [3] Latent Visual Reasoning

---

> > ### Comment · Reviewer_eNuB · 2025-11-27
> > **Response by Reviewer**
> >
> > Thank you for addressing my concerns. I am especially glad to see the latency results, as well as your detailed thoughts on how you interpret “perception” and “reasoning.”
> >
> > I will raise my original score to 6.

---

> > > ### Author Response · Authors · 2025-11-27
> > >
> > > We truly appreciate your recognition of our work! Your suggestions and the experimental insights you shared have been invaluable in strengthening the robustness of our paper. Thank you for your dedicated review and constructive feedback.

---

### Official Review · Reviewer_fJsx · 2025-11-01

**Soundness:** 3
**Presentation:** 3
**Contribution:** 2
**Rating:** 4
**Confidence:** 5

**Summary:**

This paper investigates whether inference-time scaling techniques can enhance the visual perception capabilities of Large Vision-Language Models (LVLMs). The authors introduce DisTANCE, a perception-centric benchmark focused on visual estimation tasks involving geometric shapes. Their evaluation reveals that current LVLMs exhibit limited estimation precision, with inference-time scaling offering only marginal improvements. To address this limitation, the authors propose Perception-Time Scaling (PTS), a novel paradigm that encourages token-rich perception through two key components: (1) Perception Elaboration, which uses symbolic tokens (e.g., <==========>) to represent abstract visual attributes like distance, and (2) Perception Decomposition, which breaks down complex perception tasks into tractable sub-problems. The method is trained via a two-stage pipeline: supervised fine-tuning followed by GRPO-based reinforcement learning.

**Strengths:**

I appreciate the core insight of this paper that the observation about "fast perception" in Section 2.3 is quite revealing: reasoning models produce longer chains of thought, but the perception-related content remains sparse. This is an important finding that hasn't been explicitly studied before, and the quantitative analysis of perception ratios in Table 2 makes this concrete.

What I find most interesting is the symbolic tokenization idea. Using <==========> to represent distances feels almost toy-like at first, but it's actually quite elegant. It gives the model a way to "think through" visual measurements step by step, similar to how humans might count or measure incrementally.

The paper is generally well-written. Figure 1 effectively sets up the problem, and Figure 3 clearly illustrates the method. The examples in the appendix (Figures 7-9) are helpful for understanding how the approach works in practice. I can see the step-by-step decomposition and symbolic representation clearly.

**Weaknesses:**

1. Narrow task selection: While DisTANCE is well-designed for its purpose, the evaluation is limited to very specific geometric estimation tasks. The paper would benefit from demonstrating the method's effectiveness on a broader range of perception-centric tasks beyond length, perimeter, and area estimation of synthetic shapes.

2. For out-of-domain evaluation (Table 4), the paper only reports results on specific subsets like Comparison subset from Geoperception and the Height subset from LEGO-Puzzles. Both benchmarks have multiple other subsets that test different aspects of spatial reasoning. What is the performance on these other subsets? Selective reporting raises concerns about cherry-picking results.

3. In Table 5, only three subsets of BLINK (Jigsaw, Multi, Local) are reported. BLINK contains many more subsets designed to evaluate different perceptual capabilities. Why were only these three selected? A comprehensive evaluation across all BLINK subsets would strengthen the claims about perception improvement.

4. Table 5 introduces a different training setup by combining PTS data with Geometry3K math reasoning data. While this shows that PTS can be combined with other data, it doesn't directly address whether the core PTS training (SFT + RL on synthetic data) improves general multimodal capabilities. It would be more convincing to see results of the DisTANCE-trained model (from Table 3) evaluated on the general benchmarks in Table 5 to understand whether the perception improvements truly generalize.

5. Many related works misses several recent important works on multimodal reasoning and visual perception enhancement:
- VL-Rethinker (https://arxiv.org/abs/2504.08837): Addresses similar issues in visual reasoning
- ReVPT (https://arxiv.org/abs/2509.01656): Relevant work on vision-language model perception
- R-4B (https://arxiv.org/abs/2508.21113): Related to multimodal reasoning enhancement
- Zero-Thinker (https://arxiv.org/abs/2503.05132): Relevant to inference-time reasoning approaches

**Questions:**

1. The paper commits to a specific symbolic representation (<==========>) but doesn't explore whether other symbolic forms might work equally well or better. For instance:
- Would using numbers with special tokens (e.g., [DIST:1.0], [DIST:0.1]) be more efficient?
- Could using other symbols or patterns provide similar or better results?
- How sensitive is the approach to the granularity of the symbolic representation beyond the δ=0.1 vs δ=0.05 comparison?

2. While Table 5 shows some transfer to real-world perception tasks, the mechanism of this transfer is not well understood. Why would training on synthetic geometric shapes improve performance on natural images with complex visual content?

3. The symbolic representation significantly increases sequence length. For a distance of 10 units, the model would generate 100+ tokens (<==========> repeated 10 times). What are the computational costs? How does this scale?

4. What are the main failure modes of PTS? Are there specific types of geometric configurations or estimation tasks where the approach struggles? Understanding limitations would be valuable.

---

> ### Author Response · Authors · 2025-11-23
> **Response to Reviewer fJsx [1/2]**
>
> Thank you for your thoughtful feedback and insightful comments. We truly appreciate the time you took to review our paper. Below, we address each of your points in detail. We hope these responses help clarify our paper and address your concerns. Additionally, we have made corresponding changes in the PDF, with relevant sections highlighted in *light blue*.
>
> > Q1: Could the paper demonstrate the method's effectiveness on a broader range of perception-centric tasks beyond geometric estimation of synthetic shapes?
>
> We believe that PTS enhances the accurate distance estimation capabilities of LVLMs, which is a foundational and atomic perception ability that many complex VQA tasks rely on. For example, in chart understanding tasks, it is often necessary to assess the relative size relationships between different legends (e.g., the height of bars in a bar chart) to answer questions. PTS allows models to perform these tasks more accurately. Another common scenario involves understanding top-down views (e.g., floor plans of rooms), where a significant amount of spatial information is involved. Additionally, as the model's perception of distance improves, it can better distinguish subtle differences between objects. To highlight these improvements, we selected several representative cases from benchmarks such as MM-Vet and HallusionBench, which are presented in the Appendix.
>
> >  Q2: What is the performance on the full set of Geoperception and LEGO-Puzzles benchmark?
>
> For the evaluation of the Geo and Lego tasks, we specifically selected the LineComparison and Height subsets because these subsets focus primarily on distance estimation across different scenarios, which are directly related to the visual estimation tasks involved in PTS. Therefore, we chose them to test the model's ability to generalize to out-of-distribution estimation tasks. Now, we include the performance results of both the base model and the model trained with PTS on the full dataset:
>
> | Model                | Geo_full | LEGO_full |
> | -------------------- | -------- | --------- |
> | Qwen2.5-VL-7B        | 20.65    | 21.6      |
> | Qwen2.5-VL-7B-PTS-RL | 21.33    | 22.3      |
>
> As seen, our method outperforms the baseline across the entire dataset.
>
> > Q3: What is the performance on the full set of BLINK?
>
> Due to time constraints, we initially focused on subsets more directly related to distance estimation, such as jigsaw and object localization. We have now evaluated the full set, and the results are included in the table below as well as in Table 5.
>
> | Model        | BLINK |
> | ------------ | ----- |
> | Qwen2.5-VL   | 55.0  |
> | + Geo3K      | 54.3  |
> | + PTS, Geo3K | 55.9  |
>
> As shown, PTS continues to outperform the baseline across the full dataset, further demonstrating the generalization ability of our method.
>
> > Q4: What is the performance of DisTANCE-trained model (from Table 3)  on the general benchmarks in Table 5?
>
> We evaluated the models from Table 3 (SFT and RL with PTS data only) on several general benchmarks in Table 5. The results are as follows:
>
> | Model      | MMVet | HallusionBench | MMBench |
> | ---------- | ----- | -------------- | ------- |
> | Qwen2.5-VL | 66.93 | 51.2           | 83.28   |
> | PTS-only   | 69.27 | 49.4           | 81.22   |
>
> As seen, the PTS-only model shows better performance on MM-Vet and slightly lower performance on the other two benchmarks.
>
> This outcome aligns with our expectations. The experiments in Table 3 were designed to fully explore the potential of the PTS paradigm in enhancing model performance. For evaluating how PTS improves general perception abilities, we have reserved those experiments for Table 5.
>
> It's important to note that our goal was never to achieve generalization across all tasks solely by training the model on abstract estimation tasks. Instead, we aim to introduce the concept that "perception can be integrated into the reasoning process to enhance it." To demonstrate this, we introduce visual estimation tasks which focus on length estimation—a simple visual feature that can be easily represented with symbolic tokens. This characteristic allows us to integrate perception into the reasoning process (i.e., using symbolic tokens like `<==...==>`) at a relatively low cost, thereby validating the core idea of our approach.
>
> Regarding the generalization ability of our method, as shown in Table 5, mixing PTS data with other datasets improves the model's generalization. This demonstrates that integrating perception into reasoning not only improves task-specific performance but also enhances the model's overall perceptual capacity. Our attention analysis in Figure 4 further supports this improvement.

---

> ### Author Response · Authors · 2025-11-23
> **Response to Reviewer fJsx [2/2]**
>
> > Q5: More ablation on the design of symbolic tokens and representation granularity.
>
> Thank you for your suggestion. We have conducted more detailed ablation tests on the design of symbolic tokens.
>
> 1. symbolic token style
>
>    We explore various symbolic token styles. Following your suggestion of combining numbers with special tokens, we replace a line segment's symbolic representation with `[Dist: xxx]`. We also test other abstract tokens, such as `#` ，which are closer to length units, and `?` and  `*`, which is unrelated to length. We replace the symbolic tokens in the original SFT data and performed the same SFT training. The results are shown in the table below and the PDF.
>
>    | Symbolic token | RA_0.1 | RA_avg |
>    | -------------- | ------ | ------ |
>    | =              | 16.3   | 45.4   |
>    | #              | 16.7   | 42.9   |
>    | *              | 13.4   | 40.7   |
>    | ?              | 13.0   | 39.6   |
>    | [DIST:xxx]     | 15.7   | 42.0   |
>
>    The results show that `[DIST:xxx]` performs relatively poorly compared to the original PTS data. This indicates that perception decomposition alone is not enough; perception elaboration is also necessary for more accurate estimation. For other symbolic tokens, `#` performed the best, while `*` and `?` performed worse. We believe this is because the shapes of `*` and `?` are more abstract, making it harder for the model to associate them with distances.
>
> 2. Representation granularity
>
>    In the supplementary materials, we discuss the granularity of 0.05 units per symbolic token. We have now also tested granularities of 0.2 and 1.0.
>
>    | Granularity | RA_0.1 | RA_avg |
>    | ----------- | ------ | ------ |
>    | 0.05        | 10.7   | 38.2   |
>    | 0.1         | 16.3   | 45.4   |
>    | 0.2         | 12.5   | 44.7   |
>    | 1.0         | 8.0    | 46.3   |
>
>    Notably, when the granularity is set to 0.05, the performance drop is much more significant compared to the RL version. This suggests that longer symbolic tokens increase the learning difficulty, requiring subsequent RL optimization to improve performance. On the other hand, when the granularity is too coarse, the model achieves relatively better overall estimation accuracy, but struggles with high-precision estimation, as indicated by the lower RA_0.1.
>
> > Q6: Why would training on synthetic geometric shapes improve performance on natural images with complex visual content?
>
> First, we argue that the distance estimation capability enhanced by PTS data can be seen as an atomic perception ability. While many multimodal tasks may not directly involve distance estimation, precise distance judgment is crucial for solving these tasks. In the appendix, we provide additional examples illustrating how PTS data enhances performance on general questions.
>
> Second, the structured reasoning chains introduced through GRPO training are transferable across tasks. Even when the data domains do not align, these reasoning patterns can still improve the model's ability to solve complex problems. Moreover, previous research has shown that RL training on domain-specific data, such as mathematics, can also generalize well to other domains [1].
>
> > Q7: What are the computational costs of the proposed method?
>
> we design an efficiency experiment comparing the performance of Qwen2.5-VL-7B, Qwen2.5-VL-7B-CoT, RL-trained Qwen2.5-VL-7B-PTS, and tool-augmented ReVPT-7B (post-trained from Qwen2.5-VL-7B) on the DisTANCE dataset. We focused on two metrics: average sequence length and inference latency. The results are shown in the table below and in the PDF.
>
> | Model             | Sequence Length | Inference Latency |
> | ----------------- | --------------- | ----------------- |
> | Qwen2.5-VL-7B     | 314.0           | 2.71              |
> | Qwen2.5-VL-7B-CoT | 333.1           | 2.85              |
> | ReVPT-7B          | 360.5           | 6.23              |
> | Qwen2.5-VL-7B-PTS | 571.8           | 4.87              |
>
> As shown in the table, although the PTS method generates longer sequences, the increase in inference latency is still smaller compared to the tool-based method, highlighting the computational efficiency of our approach.
>
> > Q8: What are the main failure modes of PTS? Are there specific types of geometric configurations or estimation tasks where the approach struggles?
>
> After carefully reviewing the cases generated by PTS in the DisTANCE dataset, we did not observe clear failure modes (e.g., specific geometry shapes). The distribution of ground-truth values also have limited impact on the final relative error.
>
> > Q9: Missing related work.
>
> Thank you for pointing this out. Adding a discussion and comparison with these works would indeed enhance the quality of our paper. We have now included references to these works in the Related work section and further evaluated the performance of these models on DisTANCE in Table 1.
>
> **Reference**
>
> [1] NoisyRollout: Reinforcing Visual Reasoning with Data Augmentation
>
> ##

---

> > ### Comment · Reviewer_fJsx · 2025-11-27
> >
> > Thanks for your rebuttal. Although it seems that PTS does not generalize well on general benchmarks (MMbench and Hallucionbench) and suffers from sequence length efficiency problem, it can still bring insights to community and advancement in building perception-centric models. I am glad to raise my score to 6.

---

> > > ### Author Response · Authors · 2025-11-28
> > >
> > > Thank you for your decision! Your review and suggestions really help us to refine our work. We will continue to focus on developing more general, perception-centric approaches in this field.

---

### Official Review · Reviewer_J9pJ · 2025-11-01

**Soundness:** 3
**Presentation:** 3
**Contribution:** 3
**Rating:** 6
**Confidence:** 3

**Summary:**

This paper studies the effectiveness of inference time scaling for perception-centric tasks. It first proposes a new visual estimation benchmark for testing LVLMs' perception capability and then proposes the idea of Perception Time Scaling, which encourages the model to produce richer perception-related tokens in perception-heavy tasks. Experiment results show promising performance improvements.

**Strengths:**

1. I like the idea of introducing the DisTANCE benchmark, which focuses on testing the perception of LVLMs. The idea is simple yet probes the models' capability in perception, in isolation of their textual reasoning capabilities. Experiment results show existing models struggle on this benchmark, even for the proprietary ones, and that reasoning models provide only marginal gains.

2. Analysis on the ratio of perception-related tokens (in Table2 and Figure 2) provides insights and good intuitions on the proposed method on perception elaboration and perception decomposition as the main axes for inference time scaling.

3. Main experiment setup ablates the effect of data (including Direct and CoT) and training objectives (SFT and RL). Evaluations cover perception related tasks, and also general multimodal benchmarks. The results are promising that PTS improves quite significantly on visual estimation tasks and maintain performance on general multimodal tasks.

**Weaknesses:**

1.  While the high-level idea of perception time scaling is interesting and perhaps general, the proposed method appears to be quite task-specific for visual estimation kind of tasks. For example, in perception elaboration, it is specifically using <==========> to represent length, and the decomposition part that teaches the models to use the operation of tiling both appears to be specific for visual estimation. Have the authors thought about how this perception time scaling paradigm can be used for other perception-centric tasks?

2. In the experiments, it is perhaps less surprising to see improvements on the proposed DisTANCE benchmark (given that the models are trained on this specific dataset). My main concern is how general the proposed method is. For instance, we see relatively much less performance improvements when tested on OOD datasets comparing PTS and the baselines. For Geo and Lego tasks, why are the other subsets not considered?

**Questions:**

1. In the experiments, why is there no "Direct with SFT" as a baseline?

Please see other questions in the weakness section.

---

> ### Author Response · Authors · 2025-11-23
> **Response to Reviewer J9pJ**
>
> Thank you for your positive feedback and insightful comments! We truly appreciate the time you took to review our paper. Below, we address each of your points in detail. We hope these responses help clarify our paper and address your concerns. Additionally, we have made corresponding changes in the PDF, with relevant sections highlighted in *light yellow*.
>
> > Q1: Have the authors considered how the perception time scaling paradigm can be applied to other perception-centric tasks beyond visual estimation?
>
> We believe that PTS enhances the accurate distance estimation capabilities of LVLMs, which is a foundational and atomic perception ability that many complex VQA tasks rely on. For example, in chart understanding tasks, it is often necessary to assess the relative size relationships between different legends (e.g., the height of bars in a bar chart) to answer questions. PTS allows models to perform these tasks more accurately. Another common scenario involves understanding top-down views (e.g., floor plans of rooms), where a significant amount of spatial information is involved. Additionally, as the model's perception of distance improves, it can better distinguish subtle differences between objects. To highlight these improvements, we selected several representative cases from benchmarks such as MM-Vet and HallusionBench, which are presented in the Appendix.
>
> > Q2: For Geo and Lego tasks, why are the other subsets not considered? How general is the proposed method?
>
> For the evaluation of the Geo and Lego tasks, we specifically selected the LineComparison and Height subsets because these subsets focus primarily on distance estimation across different scenarios, which are directly related to the visual estimation tasks involved in PTS. Therefore, we chose them to test the model's ability to generalize to out-of-distribution estimation tasks. Now, we include the performance results of both the base model and the model trained with PTS on the full dataset:
>
> | Model                | Geo_full | LEGO_full |
> | -------------------- | -------- | --------- |
> | Qwen2.5-VL-7B        | 20.65    | 21.6      |
> | Qwen2.5-VL-7B-PTS-RL | 21.33    | 22.3      |
>
> As seen, our method outperforms the baseline across the entire dataset.
>
> As for the generability of PTS, we conducted additional experiments in Table 5, where we mixed our PTS data with other general multimodal data for training. The results demonstrate that PTS data effectively enhances the model's performance on general perception-centric tasks.
>
> > Q3: Why is there no "Direct with SFT" as a baseline?
>
> Our original intention was to compare the performance of different patterns of reasoning chains, so we take the CoT SFT as the baseline. We have now added the performance of Direct SFT in the below table and Table 3 in the PDF.
>
> | Model               | Length | Perimeter | Area | Average |
> | ------------------- | ------ | --------- | ---- | ------- |
> | Qwen2.5-3b + Direct | 44.3   | 57.8      | 30.2 | 44.1    |
> | Qwen2.5-3b + CoT    | 48.0   | 61.8      | 34.2 | 48.0    |
> | Qwen2.5-7b + Direct | 49.4   | 63.0      | 33.4 | 48.6    |
> | Qwen2.5-7b + CoT    | 40.8   | 52.2      | 29.2 | 40.7    |
>
> The results show that Direct SFT have comparable performance with CoT SFT, demonstrating that models can not easily benefit from reasoning-centric data on perception-centric tasks.

---

### Author Response · Authors · 2025-11-27
**Follow-Up on Rebuttal and Discussion**

Dear Reviewers,


We hope this message finds you well. As the discussion period comes to a close, we would like to kindly draw your attention to our previous rebuttal.


We would greatly appreciate your feedback on whether our responses have sufficiently addressed the concerns you raised. We remain eager to hear about any remaining issues or additional suggestions that could help us further improve our work.


Thank you once again for your thoughtful feedback and the constructive discussion.


Best regards,

The Authors

---

### Author Response · Authors · 2025-12-02
**Summary of Our Responses and Revisions during the Rebuttal Period**

Dear Area Chairs,

We sincerely thank you for your time and effort in handling our paper. We have summarized the feedback from the review stage to assist in your judgment.

Our work aims to enhance the perception capabilities of LVLMs by addressing the limitations of current fast perception paradigms. We propose Perception-Time Scaling (PTS), a novel inference paradigm for LVLMs that integrates the perception process into the reasoning process.

---

# 1. Reviewer Feedback

Our initial scores were **8 (Reviewer EEJH), 6 (Reviewer J9pJ), 4 (Reviewer eNuB), 4 (Reviewer fJsx)**.

During the rebuttal period, our responses and revisions were positively acknowledged by all reviewers:

- Reviewer EEJH and Reviewer J9pJ **maintained their positive scores**.
- Reviewer eNuB appreciated our detailed responses, particularly regarding our discussion on integrating "perception" with "reasoning." **Their score was raised to 6**, with confidence 4.
- Reviewer fJsx confirmed that their main concerns were resolved and acknowledged that our work provides valuable insights into building perception-centric models. **Their score was raised to 6**, with confidence 5.

In summary, after the rebuttal, our final scores were **8 (Reviewer EEJH), 6 (Reviewer J9pJ), 6 (Reviewer eNuB), 6 (Reviewer fJsx)**.

---

# 2. Our Strengths

We are truly encouraged that the reviewers acknowledged the contributions and strengths of our work, which we briefly summarize below:

- **Novelty of the Approach**: Reviewers J9pJ and eNuB praised the novelty of the DisTANCE benchmark, highlighting how it isolates perception tasks and reveals challenges faced by current LVLMs. Reviewer fJsx also appreciated the novel insight in Section 2.3, noting that while reasoning models generate longer chains of thought, perception-related content remains sparse.
- **Symbolic Tokenization**: Reviewer fJsx appreciated our symbolic tokenization approach, particularly the use of "<=========>" for representing distances, finding it elegant and akin to human reasoning.
- **Perception-Centric Benchmark and Methodology**: Reviewer eNuB recognized the introduction of the Perception-Time Scaling (PTS) paradigm, which enhances smaller VLMs' perception capabilities. Reviewer J9pJ appreciated our analysis of perception-related tokens, highlighting their role in inference-time scaling.
- **Comprehensive Experiments**: All reviewers acknowledged the rigorous experimental setup, including ablations on data and training objectives. Reviewers EEJH and J9pJ noted that PTS significantly improved performance on visual estimation tasks and maintained robustness on general multimodal benchmarks. Reviewer EEJH specifically praised the CoT vs. PTS ablation, where PTS showed improvements exceeding 10 accuracy points on DisTANCE tasks.
- **Clarity and Presentation**: Reviewer fJsx found the paper well-written and clearly structured, with figures like Figure 1 and 3 effectively illustrating the method. The appendix examples (Figures 7-9) further clarified the approach.

---

# 3. Main Responses and Revisions

We appreciate the insightful comments and suggestions from the reviewers and have provided additional experimental results, clarifications, and manuscript revisions during the rebuttal phase. Below we summarize our responses and revisions:

- **Experiments**
  - Effect of scaling training data: Tested the effect of using more training data on the model's performance on DisTANCE in Section 4.4 (reviewer eNuB).
  - Discussion on reward format: Discussed discrete and continuous reward formats and compared the performance of both in Section 4.4 (reviewer eNuB).
  - Computational costs: Provided efficiency experiments demonstrating that our method has lower computational costs and better performance compared to tool-based methods in Section 4.4 (Reviewers fJsx, eNuB).
  - Effect of prompting strategies: Evaluated the in-context learning baseline in Appendix D (reviewer eNuB).
  - More ablation studies: Added more ablation studies on symbolic token style and granularity in Appendix E (reviewer fJsx).
  - Additional baselines: Added results for direct SFT in Table 3 (Reviewer J9pJ).
- **Discussion & Clarification**
  - Provided a detailed discussion about potential directions for the integration of perception and reasoning.
  - Related work: Added results for more open-source LVLMs (Reviewer fJsx) and closed-source LVLMs (Reviewer eNuB) in Table 1, with further discussion in Appendix L.
- **More Qualitative Cases**
  - Added more cases where PTS helps the model perform general tasks in Appendix K (Reviewers J9pJ, fJsx).

---

We believe these revisions have strengthened the technical rigor, clarity, and completeness of our work. **All changes have been incorporated into the revised version, with modifications clearly highlighted**.

We hope this summary clarifies the review process and assists in your decision. Thank you once again for your valuable time and consideration.

Best regards,

Authors

---

### Meta-Review · Area_Chair_P2gx · 2026-01-04

**Summary:**

This paper investigates the limits of inference-time scaling for perception-centric tasks in Large Vision-Language Models (LVLMs). The authors introduce DisTANCE, a perception-focused benchmark for fine-grained visual estimation, and propose Perception-Time Scaling (PTS), a paradigm that integrates perception elaboration and decomposition into the inference process using symbolic tokens and reinforcement learning.

Reviewers generally agreed that the paper identifies an important and underexplored limitation of current LVLMs –– namely, their tendency toward fast, shallow perception –– and provides a technically sound and well-motivated solution. While initial reviews were mixed, the rebuttal substantially clarified methodological choices, expanded experiments, and addressed concerns about generality, efficiency, and evaluation scope. Overall, the paper was viewed as a meaningful contribution to perception-centric multimodal reasoning.

**Reviewer Concerns:**

**Concerns addressed by the rebuttal:**

* **Generality beyond synthetic visual estimation:** The authors added evaluations on full versions of GeoPerception, LEGO-Puzzles, and BLINK, as well as mixed-data training experiments, demonstrating that PTS can generalise beyond the DisTANCE benchmark.
* **Selective reporting & missing baselines:** Additional baselines (e.g., Direct SFT), full benchmark results, and expanded ablation studies on symbolic token design, granularity, and reward formulation were provided.
* **Efficiency and scalability:** New latency and sequence-length analyses showed that PTS, while producing longer sequences, remains more efficient than tool-augmented alternatives.
* **Reward and training design:** The authors justified the use of continuous rewards and supported this choice with further ablations.
* **Clarity and framing:** The relationship between perception and reasoning, and the motivation behind the term “Perception-Time Scaling,” were clarified with additional discussion.

**Concerns partially remaining:**

* While PTS clearly improves perception-heavy tasks, gains on broad, general-purpose multimodal benchmarks remain modest, and the approach appears most impactful for measurement-like perception tasks. Some reviewers noted that the scope of applicability beyond such tasks remains an open question.
* The symbolic design, though effective, may not yet represent a fully general mechanism for perception across diverse visual domains.

**Reviewer Scores:**

Reviewer EEJH: Maintained a strong positive evaluation throughout (final score 8).

Reviewer J9pJ: Maintained a marginally positive score (6), with concerns about generality largely addressed.

Reviewer eNuB: Raised score from 4 to 6 after rebuttal, citing satisfaction with added experiments, efficiency analysis, and conceptual clarification.

Reviewer fJsx: Raised score from 4 to 6 after rebuttal, acknowledging remaining limitations in generalisation but recognising the paper’s insights and contributions.

---

### Decision · Program_Chairs · 2026-01-26

Accept (Poster)